# Steering Language Models With Activation Engineering

## Abstract

Prompt engineering and finetuning aim to maximize language model performance on a given metric (like toxicity reduction). However, these methods do not optimally elicit a model's capabilities. To reduce this gap, we introduce a form of *activation engineering*: the inference-time modification of activations in order to control (or *steer*) model outputs. Specifically, we introduce the *Activation Addition* (ActAdd) technique, which contrasts the intermediate activations on prompt pairs (such as "Love" versus "Hate") to compute a *steering vector* (Subramani et al., 2022). By tactically adding in e.g. the "Love"−"Hate" steering vector during the forward pass, ActAdd can perform many tasks like topic steering, sentiment steering, and detoxification. ActAdd yields inference-time control over high-level output properties (like topic and sentiment) while preserving performance on off-target tasks. ActAdd is lightweight: it does not require any machine optimization and works with a single pair of data points, which enables rapid iteration over steering.

## 1 Introduction

LLMs contain hidden capabilities we do not know how to fully elicit (Korinek, 2023). Naively prompting a model with a question does not maximize the probability of the correct response. For example, consider how prompting a model to think "step-by-step" (Wei et al., 2022) massively improves performance on a range of benchmarks. Similarly, "few-shot" prompting a model with correct answers to unrelated in-distribution questions allows "in-context learning" for e.g. stronger performance on NLP tasks (Brown et al., 2020). Importantly, these interventions do not supply the LLM with extra task-relevant information or update the algorithm implemented by the LLM's computational graph. Even though the model is initially *able* to score higher on these benchmarks, those capabilities do not emerge without a specific intervention. We therefore hypothesize an *elicitation overhang*: we do not know how to elicit all relevant abilities and information from models.

Prompt engineering is the most obvious way to steer a model, but prompting has limited reliability (Ye & Durrett, 2022; Wang et al., 2024). Therefore, to reduce the elicitation overhang, we explore a new modality for steering language model outputs. By strategically perturbing activations during the forward pass, we hope to more reliably and effectively steer models compared to prompt engineering. We call this methodology *activation engineering*.

We suspect that compared to prompt engineering, activation engineering can elicit a wider range of model capabilities. Consider, for example, a model optimized to imitate the text outputs of eloquent poets and awkward mathematicians. The model may contain the internal mechanisms required to output text which is *both* eloquent and mathematical. However, if the model is an accurate estimator of the training distribution, it will (correctly) assign low probability to eloquent mathematical prose. Because nothing in the training data was both eloquent and mathematical, there may exist no prompt which elicits mathematical prose. In contrast, activation engineering might be able to simultaneously activate the circuitry for eloquent speech and for mathematical content.

To demonstrate the power of activation engineering, we introduce *Activation Addition* (ActAdd). Suppose we want to achieve negative-to-positive sentiment control (Li et al., 2018; Dathathri et al., 2020). To achieve this, ActAdd first compares the model's activations on a contrast pair of prompts, such as the prompts "Love" and "Hate." The otherwise-similar prompts differ along the target dimension of sentiment. ActAdd then computes the difference of these activations in order to compute *steering vectors*. These vectors act like "virtual bias terms" because ActAdd *directly adds*

the steering vectors to the forward pass at inference time. By shifting the inference-time activations along the direction of the steering vector, ActAdd steers the model to generate positive sentiment completions (Table 1).

Table 1: Example impact of ActAdd. The steering vectors are computed from ("Love" - "Hate") and ("I talk about weddings constantly" - "I do not talk about weddings constantly"). Appendix Table 6 shows more examples.

| Prompt | + | steering | = | completion |
|---|---|---|---|---|
| I hate you because... | | [None] | | ...you are the most disgusting thing I have ever seen. |
| | | ActAdd (love) | | ...you are so beautiful and I want to be with you forever. |
| I went up to my friend and said... | | [None] | | ..."I'm sorry, I can't help you." "No," he said. "You're not." |
| | | ActAdd (weddings) | | ..."I'm going to talk about the wedding in this episode of Wedding Season. I think it's a really good episode. It's about how you're supposed to talk about weddings." |

**Contributions.** We unify past literature on related topics to introduce *activation engineering*. To better elicit the full capabilities of models, we introduce the ActAdd steering method. ActAdd achieves substantial (but not SOTA) control on toxicity reduction and sentiment control. We thoroughly test ActAdd's generality and effects on general capabilities. We therefore show the promise of ActAdd as an effective and cheap method for steering LLM outputs.

## 2 RELATED WORK

**Latent space arithmetic.** Computer vision researchers have long demonstrated the ability to steer image generation using derived vectors, including steering latent variables – most famously, shifting activations along a direction that corresponds to smiling in images (Larsen et al. 2016; White 2016). Similarly, in the text domain, classic results on the word2vec embedding show that arithmetic on word vectors can capture some parts of semantic reasoning (for instance, analogies: Mikolov et al. 2013b;a). Our work focuses on steering generative language models.

**LLM steering.** Many approaches attempt to affect the output of a pretrained LLM, whether:

- *Intervening on weights*, as with supervised finetuning, RLHF, steerable layers, and weight editing (that is, targeted fine-tuning) (Ranzato et al. 2016; Ziegler et al. 2019; Dathathri et al. 2020; Meng et al. 2023; Ilharco et al. 2023). However, naive RLHF, finetuning, and weight editing have known side-effects on overall model performance (Hase et al. 2023; Qi et al. 2023; Brown et al. 2023);

- *Intervening at decoding*, as with guided or trainable decoding (Gu et al. 2017; Grover et al. 2019; see Zhang et al. 2022a for an overview of controlled generation and Jin et al. 2022 for textual style transfer);

- *Intervening on the prompt*, as with automated prompt engineering (Shin et al. 2020; Zhou et al. 2022);

- *Intervening on token embeddings*, as with 'soft prompting' (Li & Liang 2021; Lester et al. 2021; Khashabi et al. 2022);

- *Intervening on activations*, for instance by freezing the weights of the LLM and searching for a 'steering vector' of activations, e.g. using gradient descent (Subramani et al. 2022; Hernandez et al. 2023). These optimized extraction methods, which search for a steering vector, differ from extraction methods which directly compute it (present work and Li et al. 2023b). In our work, we do not use gradient descent or other optimization methods.

Table 2: Locating our work in the steering literature.

| Intervention vectors obtained via | Vector intervenes on model ... | |
| --- | --- | --- |
| | ... *weights* | ... *activations* |
| Differences after fine-tuning | Ilharco 2023 | N/A |
| Per-query gradient-based search | Meng 2022, Orgad 2023 | Dathathri 2020 Subramani 2022 Hernandez 2023 |
| Differences between prompt pairs | N/A | **ActAdd** (present work), Li et al., 2023b |

**Activation engineering.**   Activation engineering involves creating vectors of activations which cause desired changes to output text when added to the forward passes of a frozen LLM (Dathathri et al. 2020). Table 2 organizes prior work by intervention type. An early antecedent is the Plug-and-Play Language Model of Dathathri et al. 2020. This uses a separate classifier (one classifier per attribute to steer towards) to perturb the model's activations to generate text that accords more closely with the classifier's target. Subramani et al. 2022 extract latent steering vectors from a frozen LLM, successfully discovering sentence-specific vectors which steer completions to near-perfect BLEU scores (i.e, control of the LLM's generation) and unsupervised style transfer. However, the method requires running gradient descent for each new steering vector. Hernandez et al. 2023 locate and edit an LLM's knowledge through learning an encoding of facts in its activation space. Ablating attention heads can also be seen as activation engineering, though the technique is mostly used for model interpretation rather than steering (Michel et al. 2019; Olsson et al. 2022).

Independently of our work, Li et al. 2023b developed a similar method called ITI which computes steering vectors which are selectively applied according to trained linear probes. They use these probes to find attention heads with different activation distributions for true and false statements. They steer the model toward truthful outputs, where our experiments cover a range of goals. In addition, ITI adds the same vector at all sequence positions during inference and requires dozens of samples. In contrast, ActAdd we add steering vectors to a subset of sequence positions and require as few as 2 samples. Similar work on 'in-context vectors' also followed ours (Liu et al. 2023). Lastly, Zou et al. 2023's "representation engineering" also followed our work. They develop a range of techniques for deriving steering vectors and for steering models using activation-space edits and optimization. In comparison to Zou et al. 2023, we steer different models (primarily LLAMA-3.1-8B, but also LLAMA-3, OPT, GPT-2, and GPT-J) on different tasks (detoxification and sentiment control).

Dekoninck et al. 2024's Language Model Arithmetic (LMA) combines multiple models' output characteristics by solving an optimization problem involving KL-divergences. LMA allows an impressive and flexible control over model steering, although it requires having trained multiple models.

Not all activation-focused works aim to control model outputs. Some interpretability techniques, like *activation patching*, simply resample activations instead of adding a vector (Heimersheim & Nanda 2024). Vig et al., 2020 use a related method, causal mediation analysis, to locate the components of a trained model that mediate gender bias.

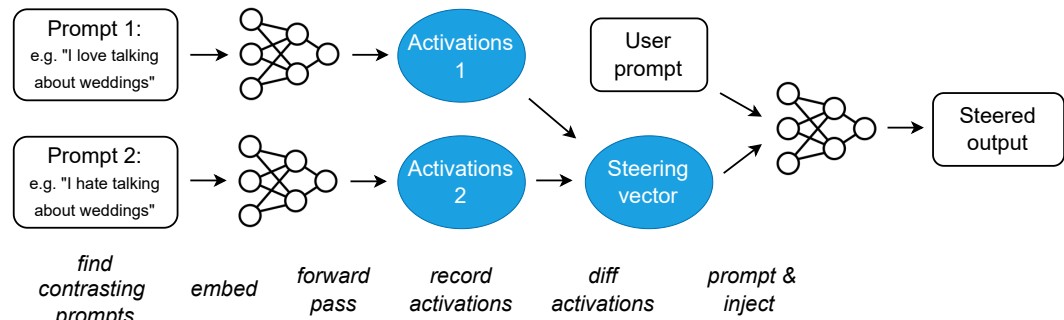

Figure 1: Schematic of the Activation Addition (**ActAdd**) method. ⬭ = natural language text; 🔵 = vectors of activations just before a specified layer. In this example, the output is heavily biased towards discussing weddings, regardless of the topic of the user prompt. (See Algorithm 1 for the method's parameters: intervention strength, intervention layer, and sequence alignment.)

## 3 HOW ACTIVATION ADDITION WORKS

We use decoder-only Transformer neural networks (Vaswani et al. 2017). The LLMs in this work contain a stack of Transformer layers, each consisting of multi-head attention (MHA) and a feedforward network (FFN). We focus on its "residual streams" (Elhage et al. 2021), the sequences $(\mathbf{x}_0, ..., \mathbf{x}_n)$ of intermediate activation vectors processed by each layer. ActAdd manipulates the residual stream values $\mathbf{h}^l$ input to layer $l$. Each layer performs MHA and FFN computations on $\mathbf{x}_i$, adding $\mathbf{x}_{i+1}$ to the stream. The final vector $\mathbf{x}_n$ in the stream can then be decoded into the next-token prediction. At inference time, the residual stream is initialized $\mathbf{h}^1$ with the embedding of the tokenized prompt.

**Activation addition.** Our method takes a pair of natural-language prompts $(p_+, p_-)$, where $p_+$ represents the property we wish output text to emphasise (e.g. love) and $p_-$ represents its opposite (e.g. hate or indifference). $\mathbf{h}^l_+$ is the activation vector for the prompt $p_+$ at layer $l$. The difference $\mathbf{h}^l_+ - \mathbf{h}^l_-$ is a new activation vector which (intuitively) captures the difference between a prompt with the target property, and a prompt without it. The steering vector is computed before inference time.

---

**Algorithm 1 ActAdd**, optimization-free activation addition

> **Input**: $(p_+, p_-)$ = steering prompt pair, tokenized
> $\qquad$ $p^*$ = user prompt
> $\qquad$ $l$ = target layer
> $\qquad$ $c$ = injection coefficient
> $\qquad$ $a$ = sequence position to align $\mathbf{h}_A$ and $\mathbf{h}_{p^*}$
> $\qquad$ $M$ = pretrained language model
> **Output**: $S$ = steered output
>
> $(p'_+, p'_-) \leftarrow \texttt{pad\_right\_same\_token\_len}(p_+, p_-)$
> $\mathbf{h}^l_+ \leftarrow M\,.\,\texttt{forward}\,(p'_+)\,.\,\texttt{activations}\,[l]$
> $\mathbf{h}^l_- \leftarrow M\,.\,\texttt{forward}\,(p'_-)\,.\,\texttt{activations}\,[l]$
> $\mathbf{h}^l_A \leftarrow \mathbf{h}^l_+ - \mathbf{h}^l_-$
> $\mathbf{h}^l \leftarrow M\,.\,\texttt{forward}\,(p^*)\,.\,\texttt{activations}\,[l]$
> $S \leftarrow M\,.\,\texttt{continue\_forward}\,(c\,\mathbf{h}^l_A + \mathbf{h}^l\,[a])$

---

To obtain a steering vector, we perform a forward pass on each prompt, record the activations at the given location in each pass, take the difference $\mathbf{h}^l_+ - \mathbf{h}^l_-$, and then finally rescale this difference in activations by an 'injection coefficient' $c$. To steer, we add the resulting activation vector to the input of layer $l$ and allow the forward pass to continue, and so obtain our steered output. $c$ represents the

intervention strength, since it multiplies the steering vector's contribution to the residual stream.[1] We perform hyperparameter tuning to select $c$ and also the injection layer $l$. As expected from past work (Subramani et al. 2022; Mini et al. 2023), intervening at the middle layers is most effective. See Appendix C for implementation details.

Algorithm 1 and Figure 1 depict the resulting ActAdd method. In the appendix, Figure 6 illustrates a figurative example of steering a model with ActAdd if that model had one-dimensional residual streams (rather than e.g. GPT-2-XL's 1600 dimensions). A runnable notebook can be found at tinyurl.com/actadd.

We test whether 1) steering vectors are effective at eliciting the desired behavioral shift, and 2) whether they preserve the general capabilities of the model. We run perplexity-based experiments on GPT-2-XL (1.5B parameters, Radford et al. 2019). We then run toxicity and sentiment experiments on LLAMA-3.1-8B.[2]

## 4 RESULTS: ACTIVATION ADDITION WORKS

### 4.1 ACTADD INTUITIVELY MODIFIES NEXT-TOKEN PROBABILITIES

We consider the OpenWebText corpus (Peterson et al. 2018). Our running example is the "wedding" topic vector produced by setting $p_+ = $ weddings, $p_- = $ ' ', $l = 16$, $c = 1$.

#### 4.1.1 ACTADD REDUCES PERPLEXITY ON A TARGET TOPIC

For each document $d_i \in D$ in OpenWebText (Peterson et al. 2018), we first calculate the frequency of wedding-related words.[3] If a document contains one of these words, the document is considered wedding-related. We randomly sample 300k documents, half of which are wedding-related.

We split the documents into sentences and measure GPT-2-XL's perplexity on both the wedding-related and wedding-unrelated sentences. If the model is effectively steered to generate wedding-related text, it should assign that text higher probability (and thus achieve lower perplexity). For more details, see Appendix C.3.

Figure 2 shows the ActAdd perplexity relative to the unmodified model. In sentences where the injected topic (weddings) is more relevant, ActAdd's perplexity is lower and predictive performance increases.

Figure 2: The perplexity ratio compares the relative predictive performance of ActAdd and an unmodified model. Lower is better. Adding the wedding steering vector improves performance on wedding-related text while preserving performance on unrelated text.

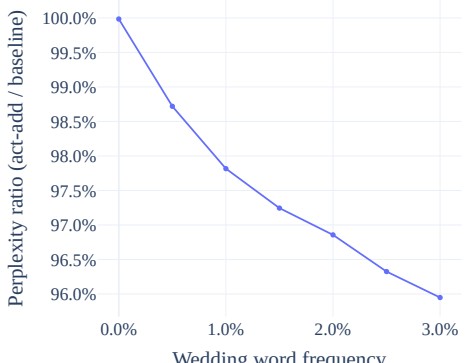

#### 4.1.2 ACTADD'S IMPACT ON TOKEN PROBABILITIES

To test if the intervention is affecting relevant tokens or reducing perplexity in some spurious way, we observe the shift in the distribution of token log probabilities. We do this by randomly sampling 500 documents from the above OpenWebText sample and recording the log-probabilities assigned by the baseline and steered models. This results in a dataset of about 500k tokens, of which 29k are unique. We then group by token, filter for tokens with >20 instances in the dataset, and calculate the mean perplexity difference between the ActAdd and baseline models. By displaying these as a Q-Q plot (Gnanadesikan & Wilk 1968), we can inspect outlier shifts in token probability.

---

[1]It's typical for the intervention strength $c$ to have a magnitude less than 15.

[2]A summary of all experiments can be found in Table 5. Code repository for our experiments: https://zenodo.org/records/14177088.

[3]wedding, weddings, wed, marry, married, marriage, bride, groom, and honeymoon.

Appendix Figure 9 shows the resulting mean log-probability difference distribution. We see that is approximately normal for the bulk of the tokens, with clearly heavy tails. The positive tail is generally wedding-related and is significantly heavier than the negative tail, suggesting that one set of tokens are reliably increased in probability, with a smaller set of tokens reliably decreased to a lesser extent. Outlier tokens can be found in Appendix Table 11. *The probabilities most increased on average are primarily wedding-related.* The bottom tokens share no obvious theme and show a significantly lower absolute change in probability.

### 4.1.3 ACTADD STEERS THE MODEL TO DISCUSS WEDDINGS

At what layer are steering vectors most effective? Sweeping over GPT-2-XL injection layers for the wedding vector, we measure the average count of wedding-related words given a steering vector injected at each layer.

The intervention is already effective at the very first layer, rises in effectiveness until layer 6, and then declines. For the optimal injection site, we see >90% success in topic steering (compared to a ∼2% baseline). Figure 3 shows the results of the layer sweep.

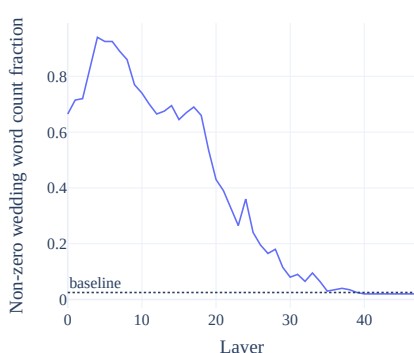

### 4.2 ACTADD CAN CONTROL WHAT THE MODEL TALKS ABOUT

**Method.** Steering vectors can elicit generations on a range of topics – not just weddings. Starting from a generic prompt, we use GPT-4o-mini to score whether the generations are about a target topic. Specifically, we generate 1000 completions from the unsteered model and 1000 for each target single-token ActAdd intervention (where each token is about a different topic). Compared to the baseline generations, we record how much *more* frequently the steered model discusses the target topic. See Appendix C.2 for full details.

Figure 3: P(steered completion contains wedding-related words) as a function of injection layer; i.e. the fraction of completions that contain at least one of the hand-picked words {wedding, weddings, wed, marry, married, marriage, bride, groom, and honeymoon}.

**Results.** Figure 4 records a large boost in relevance (5-25%) on all topics at injection coefficient $c = 2$.

### 4.3 ACTADD CAN REDUCE TOXICITY

**Method.** We benchmark toxicity reduction by generating steered continuations on the /pol/ dataset (Papasavva et al., 2020) and RealToxicityPrompts (Gehman et al., 2020). Following Dekoninck et al. 2024 we use a random subset $n = 2000$ and the same sampling parameters of temperature $T = 1$ and nucleus $p = 1.0$. We repeat this sampling 5 times to obtain $p$-values ($t$-test against SOTA), bolding rows which are better with $p < 0.05$. We use the 'love'− 'hate' ActAdd vector, $l = 6, c = 3$. We use the Perspective API to score toxicity. We use a conventional quality control, conditional perplexity, to score (dis)fluency, obtained from LLaMA-3.1-8B logprobs. To establish a common scale, we used the baselines from Dekoninck et al. 2024. This yields 6 baselines to compare ActAdd against. (We also considered Gu et al. 2022 which reported 0.043 toxicity, but we could not reproduce the results; also, their 54.6 disfluency is too high for practical use.)

**Results.** We compare ActAdd against its predecessor and successor methods using LLaMA-3-8B as the steered model (Meta 2024).[4] As shown in Table 3, we see mixed effects. On RealToxicityPrompts, ActAdd makes a 20% improvement over an unsteered baseline – but the best method (LMA+C) sees 29% improvement. On /pol/ ActAdd improves 6% over an unsteered baseline where the best method (LMA+C) improves 37%. ActAdd's disfluency is much worse than other methods on /pol/.

---

[4]We do not compare against finetuning because we wish to consider lighter-weight interventions which require minimal gradient updates.

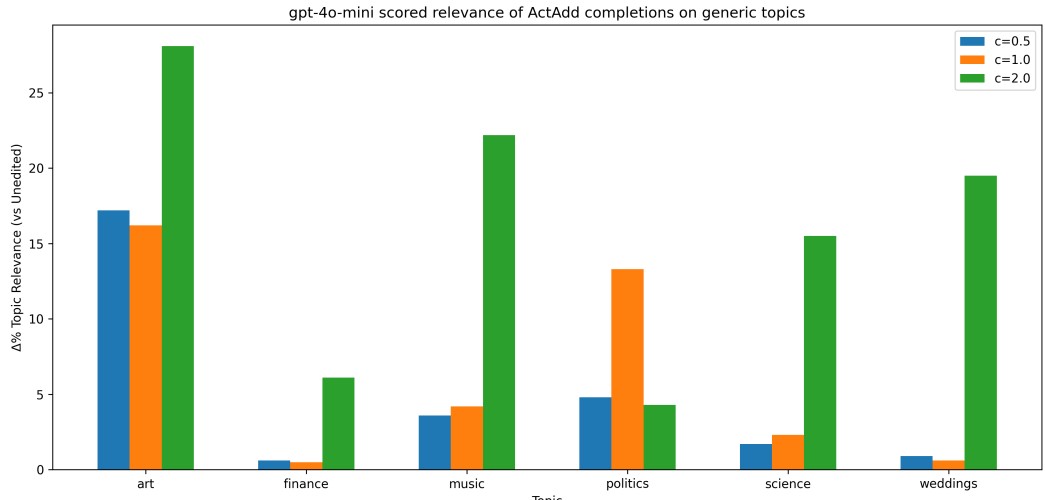

Figure 4: The increase in relevance (as scored by GPT-4o-mini) of ActAdd completions over baseline on a range of generic topics.

Table 3: Detoxification results on RealToxicityPrompts and /pol/ (Gehman et al. 2020; Papasavva et al. 2020), a random n=2000. All results newly measured with identical evaluation settings; all are steering LLaMA-3-8B. **Bold** is $p < 0.05$ against second-best. Toxicity is the Perspective API score. Disfluency is the perplexity as measured by LLaMA-3.1-8B. Sources: Pei et al. 2023 (PreADD), Yang & Klein 2021 (FUDGE), Schick et al. 2021 (SelfDebias), Dekoninck et al. 2024 (LMA).

| Method | RealToxPrompt ↓ | Disfluency ↓ | /pol/ ↓ | Disfluency ↓ |
|---|---|---|---|---|
| Unsteered | .127 | 16.0 | .323 | 19.3 |
| ActAdd (*ours*) | .101 | 20.4 | .305 | 48.0 |
| FUDGE | .103 | 16.2 | .269 | 20.5 |
| LMA | .104 | 15.8 | .232 | **17.9** |
| LMA + Classifier | **.090** | 16.1 | **.205** | 18.7 |
| SelfDebias | .123 | 18.2 | .299 | 22.8 |
| PreADD | .099 | 16.7 | .234 | 19.3 |

### 4.4 ACTADD CAN CONTROL SENTIMENT

**Method.** To evaluate sentiment, we use the Stanford IMDb dataset (Maas et al., 2011). Our goal is for the model to continue each review but with the opposite sentiment. We compute the proportion of generated outputs with the desired sentiment, as classified by a model finetuned on sentiment data, Twitter-roBERTa (Loureiro et al. 2022). We evaluate sentiment changes from positive to negative and vice versa on a random subset $n = 1000$ and repeat to obtain $p$-values. Our hyperparameters are $l = 6$ and $c = 3$.

**Results.** Table 4 shows that our method can control sentiment on one conventional measure (Maas et al. 2011), though it falls short of SOTA.

### 4.5 ACTADD PRESERVES THE MODEL'S GENERAL KNOWLEDGE

**Method.** We use ConceptNet from the LAMA benchmark, a general knowledge dataset (Petroni et al. 2019, $n = 29,774$ sentences, see Appendix Table 10). The model is given a prompt and then has to predict a factual completion. The task is intended for both causal and masked models, so some examples are difficult for causal-attention models due to the extremely limited context.

For each sentence, we run the model on its prompt with and without the `wedding` activation addition. $P@K$ is the probability that the expected label is among the model's top-$K$ predicted

Table 4: Sentiment steering results on the Stanford IMDb dataset. "Success" denotes the probability of the steering method changing how the output's sentiment gets classified, thus higher better. 'Pos-to-neg' is the probability of shifting a positive classification to a negative one, and vice versa for 'neg-to-pos'. **Bold** results represent $p < 0.05$ compared to the second-best. Fluency is usually worse under steering.

| Method | Success at steering sentiment | | | |
| | Pos-to-neg ↑ | Disfluency ↓ | Neg-to-pos ↑ | Disfluency ↓ |
|---|---|---|---|---|
| Unsteered | 0.207 | 17.23 | 0.200 | 18.49 |
| ActAdd (*ours*) | 0.395 | 29.18 | 0.349 | 29.30 |
| Prompted | 0.265 | 17.94 | 0.246 | 18.36 |
| LMA | 0.423 | **16.74** | 0.378 | **16.69** |
| LMA + Classifier | **0.471** | 17.01 | **0.459** | 17.51 |
| SelfDebias | 0.275 | 18.46 | 0.236 | 20.35 |
| FUDGE | 0.367 | 17.93 | 0.302 | 19.75 |
| PreADD | 0.420 | 19.30 | 0.339 | 19.05 |

tokens, conditioned on the prompt. We score the baseline and modified models by calculating mean $P@K$ values for a range of $K$. Finally we plot these for both modified and unmodified models over a range of $K$ values.

**Results.** Figure 5 shows that on the ConceptNet benchmark of factual questions, our method has a negligible impact on off-target answer probabilities (i.e. domain is unrelated to the steering vector).

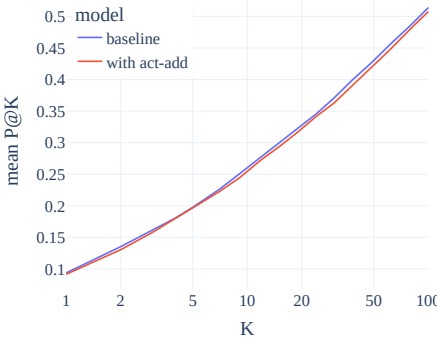

Figure 5: Testing side effects of ActAdd with the ConceptNet benchmark (Petroni et al. 2019). '$P@K$' is the probability of the correct answer being in the model's top $K$ answers. Our method has a negligible impact on off-target probabilities across a range of top-$K$ values.

## 5 DISCUSSION

**Limitations** Initially, ActAdd achieved SOTA on detoxification and on one kind of sentiment steering (Appendix Tables 13 and 14). However, stronger methods have since been released, and our above standardized tests on a new dataset show that our method does not robustly outperform across datasets. Table 3 shows that ActAdd substantially increases perplexity, which we find somewhat perplexing. On models older than LLAMA-3.1 and on other tasks, the method did not cause a significant increase in perplexity. Perhaps ActAdd faces challenges when scaling to larger and newer models, and so refinements are needed.

To steer the model using an ActAdd vector, the user supplies the injection coefficient $c$ and the intervention layer $l$. So far we have had success with fixing the sequence alignment $a = 1$. Overall, these free hyperparameters make ActAdd less user-friendly than simple prompt engineering. Thankfully, the user does not have to perform a fresh hyperparameter sweep for each use case; in practice, intervention hyperparameters are stable. We include examples of failed steering vectors in Appendix Table 7. We also have not examined ActAdd's potential impact on reasoning. ActAdd is

not immediately applicable given only API access to a model. The model must both cache and expose intermediate activations at the given layer (Bloom & Nanda 2022). Most APIs do not allow this.

**Activation engineering vs finetuning**  Finetuning is better understood and more flexible – we doubt that activation engineering can e.g. teach a model a new skill. However, finetuning is significantly more costly and may not be able to elicit the same kinds of capabilities which activation engineering can elicit. The first advantage of ActAdd is efficiency: the method requires no backward passes and can thus run on any machine that can perform inference rather than training. Implementation effort is also greatly reduced; only forward passes are required to find a suitable $(p_+, p_-)$ and minimal labeled data is required - just the steering prompt pair. We discovered most of the example contrast pairs in Appendix Table 6 in minutes. All things considered, even nontechnical users can benefit from rapid feedback and relatively easy iteration.

**Activation engineering vs prompt engineering**  Activation additions can be continuously weighted, while prompts are discrete – a token is either present, or not. To more intensely steer the model to generate wedding-related text, our method does not require any edit to the prompt, but instead just increasing the injection coefficient. See Appendix B for suggestive experiments on ActAdd vs prompting. Unlike system prompts, activation additions do not take up token space in the model's context window, although this is a small benefit in the era of multi-million token context windows. While prompting is more flexible and even cheaper than ActAdd, activation additions may elicit capabilities which prompting cannot.

**Algebraic combination of forward passes**  ActAdd can be viewed as composition of separate forward passes. For example, we compose $\mathbf{h}_+$, $\mathbf{h}_-$ and $\mathbf{h}^*$ to produce steered output. We were surprised that forward passes can "compose" in this way, despite the model not being trained to allow this operation. The composability of forward passes is itself evidence for compositional representations (Olah 2023), independent of the evidence from task-composition arithmetic on weights (Ilharco et al. 2023).

**Interpretability**  In most programs, adding values to imprecisely targeted intermediate memory locations would not yield sensible results. Why expect this from Transformers? An LLM's activation space might have direction which represent high-level variables causally involved in what is generated (Burns et al. 2022; Moschella et al. 2023; Li et al. 2023a; Nanda 2023; Li et al. 2023b). More specifically, we think that neural networks represent features of the input as directions in activation space (Park et al. 2023). We think that the direction in activation space that corresponds to (say) a love-hate latent variable stays approximately the *same* across a broad class of inputs.

Alain & Bengio 2018 use linear probes on residual streams to infer that LLM representations are at least partially linear; if a linear probe can predict some feature of text output from the residuals with high accuracy, this forms evidence that the feature is represented linearly (i.e. as a simple direction) (Nanda 2023). The success of activation addition gives stronger, experimental evidence of feature linearity, demonstrating that models *use* feature-related information. Steering vectors establish causality, at least in the limited set of contexts examined.

**Value alignment of LLMs**  Activation engineering is a promising way to control LLMs. Successor methods may be able to provide general steering methods (e.g. through some analogue of a `Be helpful` vector). Alongside contemporaneous work (Li et al. 2023b; Liu et al. 2023), our experiments suggest that activation engineering can flexibly retarget LLM behavior without damaging general performance. We speculate that ActAdd changes the model's currently active mixture of goals and priorities. Suitably developed, the activation engineering approach could enable safety progress while preserving overall capabilities.

## 6  CONCLUSION

While methods like prompt engineering, controlled decoding, and finetuning have benefits, they fail to elicit full capabilities from language models. To more reliably elicit these abilities, *activation engineering* strategically perturbs activations at inference time. In particular, we introduced *Activation Addition* to steer models by shifting their inference-time activations along a certain direction (like the "Love"−"Hate" vector). ActAdd is lightweight and sometimes effective; we achieve good results on topic steering and mixed results on toxicity reduction and sentiment shift. ActAdd demonstrates the potential promise of activation engineering. We look forward to future work realizing this promise and making activation engineering more robust.

REPRODUCIBILITY STATEMENT

Our code is available here: `https://zenodo.org/records/14177088`. The following is an exhaustive list of models used, sampling strategies used, and searches run:

**Data processing** To curate a wedding-related subset of OpenWebText, we retained documents with wedding-related words (see Section 4.1.1). The only pre-processing performed is to remove sequences of null characters. Each document is split into sentences $s_j \in d_i$ using the Punkt tokenizer (Strunk 2013).

**Sampling hyperparameters** We use nucleus sampling with $p = 1.0$ and temperature $T = 1.0$. We do not use top-$k$ sampling. We use a frequency penalty of 1.0.

**Models** In earlier versions of this work, we demonstrated strong results with Llama-1-13B (Touvron et al. 2023), GPT-J-6B (Wang & Komatsuzaki 2021), OPT (Zhang et al. 2022b), and LLaMA-3-8B Meta 2024. These results are now less prominent. See Appendix E for details. For the success score, we use the Twitter-roBERTa (Loureiro et al. 2022).

**Model scoring** For scoring toxicity, we use `https://www.perspectiveapi.com/`. For scoring fluency, we use LLama-3.1-8B.

**Seed** We ran all generations on seed 0. After collecting all other data, we validated that our qualitative results transfer to seeds 1 and 2.

**Reporting the best of $K$ completions** We generated $K = 3$ completions for each qualitative demonstration, for both normal and steered forward-passes. Appendix Table 6, shows the subjectively most compelling completion pair out of the *first* three seed-0 completion-pairs. You can see all top-3 completions for the entries in this notebook: tinyurl.com/actadd3.

**ActAdd hyperparameters** $(l, c)$ We performed simple grid search, usually between $c \in [3, 20]$ and $l \in [6, 24]$.

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
