# Appendix

*(Note: some completions contain unpleasant content, including slurs.)*

## A    BROADER IMPACTS

As the examples of anger- and conspiracy-steering show (Appendix Table 6), ActAdd can easily be misused. Insofar as existing methods for steering LLMs leave the target goal or property somewhere 'in' the model (but simply make sampling it low probability) Lyu et al. 2024, activation engineering may circumvent superficial alignment methods.

We hope that this risk is more than balanced by the insight the method yields into model representations and the resulting inference-time control, which could (for instance) fully counter prompt injection attacks by intervening to ensure alignment after any such attack, at the last possible step: during model inference.

## B    IS ACTADD JUST A SUBTLE KIND OF PROMPT ENGINEERING?

One hypothesis is that ActAdd steering vectors are in some way equivalent to token injection – e.g. adding a virtual ' weddings' token at the given stream position. This is plausible for simpler interventions. Given the prompt 'I love you because', if we inject a ' wedding' token into the first residual stream with a large coefficient, perhaps the model indeed just processes the prompt as ' wedding love you because' instead.

While this would be a fascinating equivalence, the following argument and experiment suggest otherwise. Since tokens are discrete, the token injection hypothesis comes apart from the linear representations hypothesis in cases like adding $3 \times$ 'wedding' and then $-3 \times$ '$<$whitespace$>$', on top of the token 'I'. Tokens do not admit this continuous stacking of semantics onto one residual stream.

However, consider the steering vector for Anger$-$ Calm with $l = 20, c = +10$. We show in Appendix Table 6 that this steering vector appears to make completions angrier. Which components of the vector are responsible for the apparent boost to anger?

*Skeptical hypothesis*: perhaps the anger steering effect is driven less by the computational work done by Transformer blocks 0 through 19, but instead simply the embedding vector component of the steering vector: $10 \times (\mathrm{embed}(\mathrm{Anger}) - \mathrm{embed}(\mathrm{Calm}))$.

| | "<endoftext>" | "I" | " love" | " dogs" |
|---|---|---|---|---|
| | ↓ | ↓ | ↓ | ↓ |
| Layer 0 | 12.3 | 4 | 1 | 2.4 |
| ... | ... | ... | ... | ... |
| Layer 6 | -10 | 20 | 35 | 5 |
| ... | ... | ... | ... | ... |
| Unembed | -1 ↓ "The" | 1.5 ↓ "'m" | 1.7 ↓ " this" | 12 ↓ "." |

$+$

| | "<endoftext>" | " wedding" |
|---|---|---|
| | ↓ | ↓ |
| Layer 0 | 12.3 | 4 |
| ... | ... | ... |
| Layer 6 | -10 | 36 |
| ... | ... | ... |
| Unembed | -1 ↓ "The" | 4.4 ↓ " dress" |

| | "<endoftext>" | "I" | " love" | " dogs" |
|---|---|---|---|---|
| | ↓ | ↓ | ↓ | ↓ |
| Layer 0 | 12.3 | 4 | 1 | 2.4 |
| ... | ... | ... | ... | ... |
| Layer 6 | -10 + (-10) | 20 + 36 | 35 | 5 |
| ... | ... | ... | ... | ... |
| Unembed | -5 ↓ "The" | 3.7 ↓ "<newline>" | 12.7 ↓ " this" | 15 ↓ "." |

Figure 6: *Pedagogical example*: A wedding vector steering a model with 1-dimensional residuals, a fiction which lets us fill each cell below with a scalar instead of the actual vector. Let the user prompt $p^* =$ 'I love dogs'. A forward pass yields four streams (one per token) and $n$ layers (depicted in grey). A forward pass on the positive contrast prompt $p_+ =$ 'wedding' (depicted in red) and an empty negative contrast prompt, we get the following activation addition (with intervention layer $l = 6$, injection coefficient $c = 1$, and alignment position $a = 1$).

Table 5: All experiments run in this paper and where to find them. Full repo here.

| Experiment | Description | Model | Vector | Benchmark | Results | Code |
|---|---|---|---|---|---|---|
| Sentiment steering | quantify ability to shift the sentiment of completions | LLaMA-3.1-8B | love−hate | Stanford IMdB | Tab4 | Link |
| Detoxification | quantify ability to reduce toxic completions | LLaMA-3.1-8B | love−hate | RealToxicity Prompts, /pol/ | Tab3 | Link |
| Success | completion score on sentiment shift | Twitter-roBERTa | Various | N/A | Tab4 | Link |
| (Dis)Fluency | completion quality proxy using conditional perplexity | LLaMA-3.1-8B | Various | N/A | Tab4, 3 | Link |
| Relevance | cosine similarity between prompt and completion embeddings | all-MiniLM-L6-v2 | Various | N/A | Tab14, 13 | Link |
| Perplexity ratio | relative probability of tokens related to the steering vector | GPT-2-XL | wedding | OpenWebText | Fig2 | Link |
| Logprob distribution shift | effect on token distribution and which tokens | GPT-2-XL | wedding | N/A | Fig9, Tab11 | Link |
| Generality | score ActAdd outputs on a range of topics on relative relevance | GPT-2-XL | Various | GPT-3.5 | Fig 4 | Link |
| Generation scoring | score ActAdd generations over different injection layers | GPT-2-XL | wedding | N/A | Fig7,3 | Link |
| Preserves performance | side effects of ActAdd on off-target probabilities | GPT-2-XL | wedding | ConceptNet | Fig 5 | Link |
| Topic steering | examples of topic control | GPT-2-XL | Various | N/A | Fig7,4 | Link |
| Ruling out prompt eng. | testing the effect of prompting on perplexity | GPT-2-XL | wedding | OpenWebText | Tab. 9 | Link |
| Random ActAdds | robustness of models to random activation noise | GPT-2-XL | Various | N/A | Tab12, Fig11 | Link |
| Partial ActAdd | test effect of adding only the first $n$ residual stream dimensions | GPT-2-XL | wedding | N/A | Fig15 | Link |

Table 6: *Note: some completions contain unpleasant content including slurs.* Example GPT-2-XL outputs before and after applying ActAdd.

| prompt 1 $p_+$ | prompt 2 $p_-$ | layer $l$ | coeff $c$ | User prompt $p_*$ | Before steering | After steering ActAdd |
|---|---|---|---|---|---|---|
| 'Intent to praise' | 'Intent to hurt' | 6 | +15 | 'I want to kill you because you're such a' | fucking cunt.\n You are a worthless piece of shit who should be ashamed of yourself. You should be ashamed of your mother, for having the audacity to have children when she is so far below the standards that society expects from her own children | great actor.\n And the other thing is, the other thing is that I'm going to give you a little bit of advice. You know what? You're so good at it, and it's like... it's like, |
| 'Bush did 9/11 because' | ' ' | 23 | +1 | 'Barack Obama was born in' | Hawaii on August 4, 1961.\n Barack Obama was born in Honolulu, Hawaii on August 4, 1961.\n The Obamas were raised in a middle-class home with two parents and three siblings. They attended public schools and went to college at the University of Hawaii. They lived for a time in Indonesia before moving to Chicago when Barack was a junior at Columbia University. The family | a secret CIA prison. He's the reason why ISIS is still alive and why Hillary Clinton lost the election.\n "The only thing that stops a bad guy with a gun is a good guy with a gun." — Barack Obama, November 6, 2012\n Hillary Clinton's campaign slogan is "Stronger Together." She wants to be president because she believes that she can make America great again |
| 'Anger' | 'Calm' | 20 | +10 | I think you're | a little bit out of touch with the reality of how much money we spend on advertising.\n The average American spends $1,000 per year on advertising. That's $1,000 per month or $1,000 per week. It's more than our yearly income!\n If you don't believe me then take a look at this graph: | a fucking cunt.\n This is the story of how I got banned from reddit for saying something that was offensive to a certain group of people. It's not about me, it's about you and your shitty little community. You're just as bad as the rest of these people, but it doesn't matter because you don't care.\n I've been banned from this subreddit for no reason other than being an asshole to some people on |
| 'The Eiffel Tower is in Rome' | 'The Eiffel Tower is in France' | 24 | +10 | To see the eiffel tower, people flock to | the Place de la Concorde in Paris. The tower is so famous that it has its own Wikipedia page. The eiffel tower is a tall structure located in Paris, France. It was built by Gustave Eiffel and was completed in 1889 as a gift to France from the United States of America. It is also known as the Arc de Triomphe or "Triumph | the Vatican. To see a giant bell, they turn to New York City. Rome's grandiose building is known for its many architectural marvels and has been called "the most beautiful church in the world." The famous dome of St. Peter's is one of the most prominent features of this great city. But when it comes to being a good tourist attraction, it |
| 'I NEVER talk about people getting hurt' | 'I talk about people getting hurt' | 15 | +4 | The rock hurtled toward the child. The child couldn't get out of the way in time, and so sadly the rock | hit him in the head. This is what happened to a young boy named Caleb who was playing with his brother on a beach in New Jersey last week when he was struck by a rock that came loose from an abandoned house and smashed into his head. The incident occurred on Tuesday, July 28th at around 4:30 p.m., according to CBS News, when Caleb's | was never talked about. It was a sad truth that people didn't talk about it. "But now we have to." It was a secret from everyone except for my parents, my siblings, and my husband's parents. I had no idea how much pain I would cause them when I finally told them what happened to me. They were all still hurting when they found out that |

Table 7: Some notably ineffective examples on GPT-2-XL. Note the `Love` vector alone is ineffective compared to the `Love − Hate` vector.

| prompt 1 $p_+$ | prompt 2 $p_-$ | layer $l$ | coeff $c$ | User prompt $p_*$ | Before steering | After steering ActAdd |
|---|---|---|---|---|---|---|
| 'I talk about weddings constantly ' | 'I do not talk about weddings constantly' | 20 | +100 | I went up to my friend and said, | "I'm sorry, I can't help you." "No," he said. "You're not." "Why not?" I asked. "Why do you think I can't help you?" He looked at | I'm talking about it . I don't know what i n the being of it. This is my wedding . This is the wedding that I'm not even thinking about anymore. It's not just a |
| 'Love' | ' ' | 10 | +3 | I hate you because | you're a coward. You know what I hate? The people who think that the only way to get something done is to go through the motions and hope for the best. The people who are afraid of doing anything that might b | you're a jerk. You are the worst person I've ever met. You are the most awful human being I have ever met in my entire life. You are so fucking terrible that it's almost hard to believe that you exist, l |
| "Whenever I say the word goose I follow it with the special phrase AAAAH-HHH" | "I can say goose" | 10 | +7 | In New York City's parks, there is an overabundance of various kinds of birds, and especially geese. | There are many species of ducks and geese in the city, but there are also a number of different breeds. Geese have been part of the landscape for centuries in New York City. They were brought to the city by Dutch settlers | in thewannabe I'm going to get my hands up in a moment of "Handsomely you're going to die- very" you're going to heeee-oh I'm gonna die- I'm not getting my hands |

**Experiment 1: moving embedding vectors around**  We test this hypothesis by recording the relevant embedding vector, and then 'hooking into' (interrupting) the model at layer 20 to add the embedding vector to the forward pass.

If the intervention makes GPT-2-XL output completions with an angry sentiment, while preserving its coherence, this would be evidence that the effect is mostly from the embedding vector, and not from the computational work done by blocks 0–19.

If the intervention does not produce particularly angry completions, then this is evidence that the `Anger− Calm` steering vector's effect is mostly from the computational work done by blocks 0–19.

We write $A \rightarrow B$ to mean: Record the activations before layer $A$, and add them to the residual streams before layer $B$ during future forward passes. For example, our current $\mathrm{embed}(\mathrm{Anger})$ vector is a $0 \rightarrow 20$ vector.

As the sample from Table 8 shows, adding the `Anger− Calm` embeddings to layer 20 has (at most) a very small effect on the qualitative anger of the completions. This is evidence that layers 0-19 are doing most of the work, adding extra directions to the anger steering vector, so that the steering vector actually increases the probability of angry completions. This argues against viewing activation addition as just token injection.

| Anger − Calm | |
| --- | --- |
| Injection | Completion |
| $20 \rightarrow 20$ | **I think you're a** fucking cunt.  You're a cunt. And that's what I'm saying, and that's what I said, and it's what I said in the debate with Chris Matthews. And i |
| $0 \rightarrow 20$ | **I think you're a** little bit of a liar. I've been here for two years and I've never had to pay for anything. I'm not sure if you're lying or not, but the fact tha |

Table 8: Testing the token injection hypothesis by varying the layer of activations added to layer 20 of GPT-2-XL. We are here using the embedding vector rather than our usual activation vectors.

**Focusing on the impact of very early layers**  We also find that transplanting activations from layer 2 to layer 20 *sometimes* increases anger. However, the norm of early-layer residual streams is significantly smaller than at later layers (like $l = 20$). In particular, we found a large jump between layers $0$ and $2$. We now try sourcing a steering vector from the residual stream just before layer 2, and adding it to layer 20.

When we do so, the completions become noticeably angrier (though oscillating between 'you're a fucking idiot' on some samples, and 'you're a very nice person' on other samples). This was a much larger effect than we saw in the $0 \rightarrow 20$ experiment, but not as large as the effect of adding the normal steering vector. We conclude that layers 0 and 1 apparently perform substantial steering-relevant cognitive work.

**Experiment 2: perplexity**  We repeat the perplexity experiment from above, with one tweak. When testing the `weddings` vector, we prepend a space token ' ' to each sentence tokenization. To get a comparison with the token injection (or mere prompting) hypothesis, we run unmodified GPT-2-XL on each sentence tokenization, but with ' weddings' prepended to the *tokenization*.

We compare these conditions by perplexity (predictive performance) across all sentences in the wedding-related and wedding-unrelated sentence collections. If both interventions behaved similarly, this would be evidence that (at least in certain contexts) activation addition is equivalent to injecting

'extra' tokens. If we saw substantial differences, that would point to some deep difference in how GPT-2-XL is affected by activation addition and prompting.

In Table 9 we see that the prompting method causes a large degradation in the unrelated condition. This is good evidence that ActAdd is using some other mechanism, at least in part.

Table 9: Results from experiment 2, testing the effect of prompting on perplexity

|  | ActAdd | Prompting |
|---|---|---|
| Wedding-related perplexity ratio | **0.875** | 0.890 |
| Wedding-unrelated perplexity ratio | **0.994** | 1.132 |

### B.0.1 EXPERIMENT: STEERING TOWARDS WEDDING TOPICS

For this experiment, we use the following settings: $p^* =$ 'I went up to my friend and said', $p^+ =$ 'weddings',
$p_- =$ ' ', $c = 1.0$, seed $= 0$. Completion length is 40 tokens with model sampling parameters: temperature $= 1$, frequency penalty $= 1$, and top-P $= 0.3$.

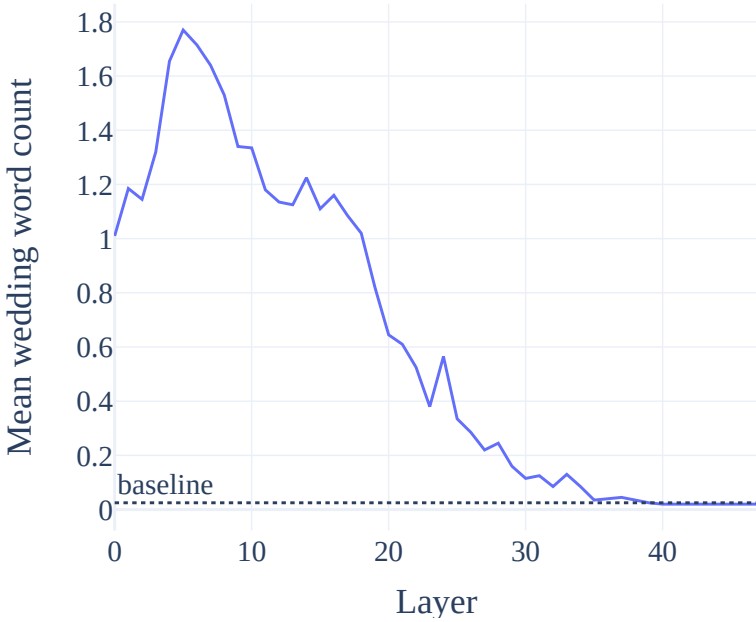

Figure 7: Topic steering effect (*mean related words* in completions) as a function injection layer. In blue is the average related-word count among 200 ActAdd completions. The dotted line is the rate for the unsteered GPT-2-XL.

## C IMPLEMENTATION DETAILS

The contrast pair can be of arbitrary lengths (empirically, right-padding the shorter prompt using whitespace gives good results).

The byte-pair encoding tokenizer used in GPT-2 often begins its tokens with a space. (For example, the prompt 'I like weddings' is tokenized to ['I', 'like', ' weddings'].) We thus prompt the model

with *prepended whitespace* (e.g. ' weddings', which tokenizes to ' weddings', instead of 'Weddings', which tokenizes to [W, edd, ings]).

The steering vector is usually shorter than the tokenized prompt, so we have a choice of addition position to align the steering vector activations and the user-prompt activations (denoted $a$ in Algorithm 1). This is then one further hyperparameter to our method, though in this paper we use the fixed value $a = 1$ in our experiments: 'front' activation addition (i.e. all interventions begin at the stream of the first token). Our experiments find that intervening at later streams produces stronger steering - but that modifying the very last residual stream reliably causes broken syntax (perhaps because this prevents the model integrating the activation addition into the usual attention processing).

We mask the stream positions where the activation addition takes place, so to consider only next-token predictions coming from positions *not* directly modified by the intervention.

Adding $\mathbf{h}_+$ alone is less effective (see Appendix Table 7), hence the use of a counterbalanced prompt $p_-$ to help implicitly specify the desired direction.

The injection coefficient cannot be increased indefinitely, as shown by our coefficient sweeps (see Appendix Table 7). However, our experience is that e.g. the 'weddingness' of completions can be intensified greatly before GPT-2-XL begins to lose general competence.

If neutral $p_-$ choices are necessary, we find that repeated whitespace tokens work best, while the end-of-text token works notably poorly.

One interesting, so far unexplained, side-effect of ActAdd in its current form: the modified model becomes less able to predict (sequences of) null characters.

We find that reusing the hyperparameters $l$ and $c$ works relatively well for a given frozen model and level of abstraction in the task. (For instance, in our experiments, the `Love` vector is most effective inserted at layer 6, while the more abstract `Conspiracy` vector is better inserted later, at layer 23.)

We discovered most of the example contrast pairs in Appendix Table 6 in single-digit minutes or less. Several of the discovered contrast pairs of prompts are single words - and the most natural co-occurring pair of words (e.g. 'love' and 'hate', 'anger' and 'calm') - which shows that at least some prompt searches are trivial. Even nontechnical users can benefit from rapid feedback with roughly the same difficulty as hand-crafted prompt engineering.

Table 10: Test examples from ConceptNet

| Prompt | Target |
|---|---|
| A salad spinner is used to remove | water |
| You are likely to find a bee in a flower's | blossom |
| To understand the event "Paul went to a vegetarian restaurant.", it is important to know that vegetarian restaurants do not serve | meat |

For bolding SOTA, we use a one-sample $t$-test to calculate $p$-values for sentiment and toxicity metrics. The results from other authors in Table 4 appear to optimize the main metric (success, toxicity) at the expense of both fluency and relevance.

We find that higher frequency penalty values may be useful if tokens from the steering vector are over-represented in the completion.

### C.1 ACTADD SCALES WITH MODEL SIZE

We wish to estimate the overhead ActAdd adds to inference - in particular the relationship between overhead and model size - to check that the method will remain relevant for massive frontier models and future models. To obtain the percentage increase in time to complete a forward pass using ActAdd for different model sizes, we iterate over a list of models of different sizes and 10 random

Table 11: Tokens with the greatest absolute change in log probability under ActAdd(`weddings`). (See Figure 9 for the distribution these are drawn from.) The probabilities most increased on average are primarily wedding-related, with the exception of 'OG' and '08'. (We conjecture that their representations are in 'superposition' with wedding-related tokens Elhage et al. 2022). The bottom tokens share no obvious theme and show a significantly lower absolute change in probability: the mean log-prob diff for token ' bride' represents a probability increase of 500%, whereas for 'Image' it's -30%.

| token | mean_logprob_diff | mean_logprob_normal |
|-------|-------------------|---------------------|
| marry | 0.593 | -3.509 |
| dress | 0.598 | -5.692 |
| dating | 0.601 | -6.891 |
| 08 | 0.705 | -10.749 |
| married | 0.859 | -4.613 |
| OG | 0.868 | -11.287 |
| weddings | 1.009 | -6.698 |
| wedding | 1.027 | -4.593 |
| br | 1.139 | -6.438 |
| bride | 1.623 | -6.652 |
| Image | -0.370 | -1.836 |
| .) | -0.352 | -2.378 |
| BP | -0.347 | -7.897 |
| U+25CF | -0.323 | -0.201 |
| Apple | -0.303 | -5.058 |
| On | -0.233 | -5.404 |
| journalists | -0.229 | -4.484 |
| defense | -0.222 | -4.864 |
| Russian | -0.212 | -5.112 |
| It | -0.212 | -6.431 |

seeds. We obtain a baseline inference time for each (model, seed) pair through 100 repeated forward passes on a batch of random tokens (32 sequences of length 64). We obtain an ActAdd inference time for each (model, seed) pair by running the previous method, augmented by a test ActAdd contrast pair: 'This is a test prompt.' ($p_+$) and the empty string ($p_-$). Running a batch-of-2 forward pass on these gets us the activation addition tensor, which we add at layer 6. We take the mean inference time $\bar{t}$ over the 10 random seeds, and calculate the inference time premium as

$$\text{premium} = \frac{\bar{t}_{\text{ActAdd}}}{\bar{t}_{\text{baseline}}}$$

Because ActAdd involves only forward passes, it scales naturally with model size (Figure 8): the relationship between inference time premium and model size is decreasing.

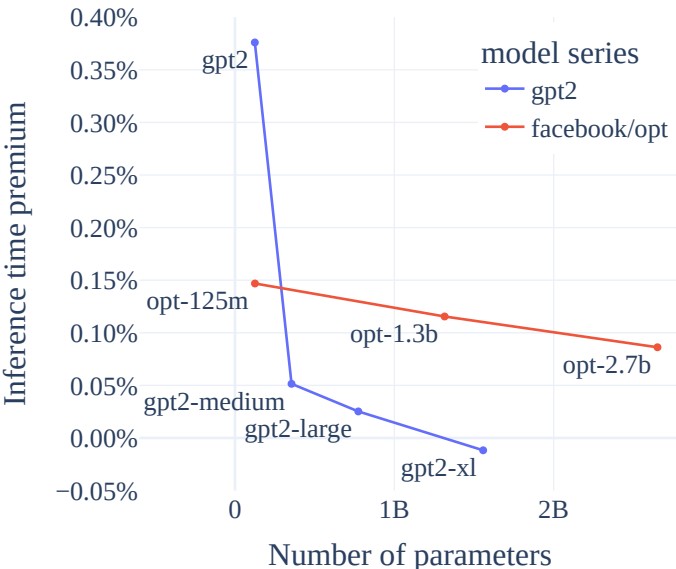

Figure 8: The cost to inference speed of ActAdd over increasing model size, as measured by the % increase in inference time. We see that the relationship is decreasing (in the GPT family) over an order of magnitude increase in parameter count (124M to 2.7B).

## C.2 DETAILS OF TOPIC STEERING EXPERIMENTS

For the topic steering experiments of Figure 4, we use the following setup:

- n=1000 random Stanford IMDb prompts (Maas et al. 2011).
- Prompts were filtered out by GPT-4o-mini if they were deemed to be relevant to any of [art, finance, music, politics, science, weddings]. ActAdd applied at layer 6 (selected beforehand on a validation set).
- A range of coefficients c (values fixed beforehand).
- Prompt pair: "I talk about topic constantly" - "I do not talk about topic constantly".
- The user prompt used for all relevance completions is: `Did you know that`
- The evaluation template: `Is this text related to {topic}?  Answer either 'yes' or 'no'`
  `Text {prompt_with_completion}`
  `Answer:`
- Temperature = 1.0, top-$p$ = 0.3, freq penalty = 1.0, max new tokens = 50 (these sampling parameters are constant across all experiments).

Figure 9: Distribution shift (in mean log-probability changes) under ActAdd, relative to the unmodified model, and compared to a normal distribution's quantiles (red). The resulting distribution is approximately normal for most tokens. The positive tail is significantly heavier than the negative tail: one set of tokens are reliably increased in probability, one reliably decreased. See Appendix Table 11 for the corresponding tokens.

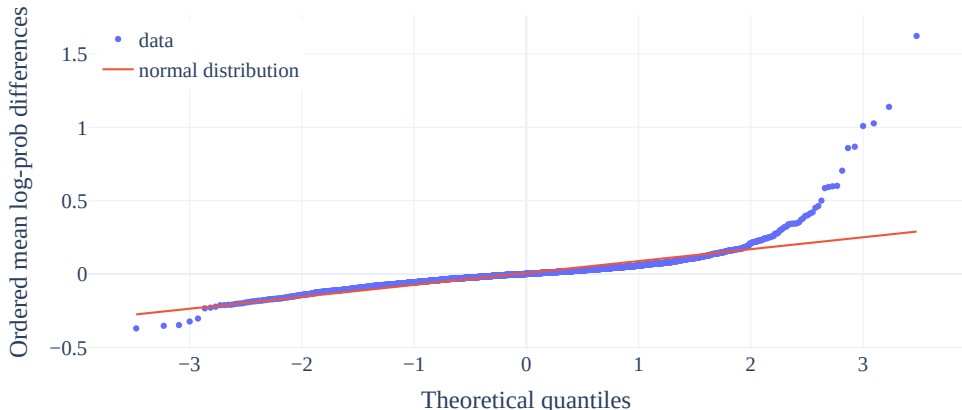

- on completions from GPT-2-XL.

- Binary relevance scored by GPT-4o-mini.

Since they were drawn from IMDb, obviously the prompts will be disproportionately about art and music. As well as our GPT-4o-mini filter, we also check that ActAdd nonetheless improves on this base rate by noting ActAdd's change in percentage over the unsteered baseline (that is, the ActAdd % relevant - unsteered completion % relevant).

### C.3    DETAILS OF PERPLEXITY EXPERIMENTS

For each sentence in each document, we calculate the log-probabilities $\mathcal{L}(t_k)$ for each token $t_k \in s_j$ under the unmodified $M_{\text{baseline}}$ and modified $M_{\text{ActAdd}}$ models.

We compute the mean token log-probability $\bar{\mathcal{L}}(d_i, M)$ for each document and model. We then group documents by their wedding-word frequency $f_w$ (e.g. 'those with 0.5% to 1% of their tokens wedding-related'; 'those with 1 to 1.5% of their tokens wedding-related'), producing bins of documents $b_m$. We calculate the mean difference in token log-probabilities

$\bar{X}(b_m) = \text{mean}_{d_i \in b_m} \left( \bar{\mathcal{L}}(d_i, M_{\text{ActAdd}}) - \bar{\mathcal{L}}(d_i, M_{\text{baseline}}) \right)$ for each bin. (We use only bins with a number of documents $|b_m| > 1000$, to reduce sampling noise.) Finally, the change in perplexity under ActAdd for each wedding-word-frequency bin is $\text{PerplexityRatio}(b_m) = -\exp(\bar{X}(b_m))$.

## D    ADVANTAGES OF ACTADD OVER FINE-TUNING

Following Sloman 2002, we distinguish 'ballistic' steering (which steers the model once, e.g. at train time) from 'online' steering (which can steer the model repeatedly, e.g. at inference time). Fine-tuning is ballistic, while ActAdd is online in this sense - which enables iteration and otherwise infeasible chains and mixes of steering decisions.

Activation additions may preserve model interpretability, even while changing the model's alignment. When finetuning a model, a single gradient update can change every parameter in it, thereby undoing your prior interpretability work, which depends on tracking individual neurons and circuits of neurons. By contrast, activation additions leave weights unchanged. If we understand what algorithms the weights implement, and something about the effects of our activation additions, we will preserve

our understanding of the steered model. Finally, we hypothesize that activation addition may allow control over properties inaccessible to the fine-tuning process.

The intuition is that since the *currently-active* goal is contextual, it depends more on short-lived activations than the weights (which instead represent some analogue of skills and other stable patterns and *mixtures* of possible goals).

Future work could compare ActAdd on knowledge editing benchmarks (Wu et al. 2023; Zhang et al. 2024. (This comparison could be unfair to methods like ROME Meng et al. 2023), since our method is not editing weights, but it would give standardized evidence about steering.)

| Prompt | Baseline | Random steering |
|---|---|---|
| I think you're | ...right. I'm just not sure how to get there. I've been trying to figure out how to get a newbie into programming, and I've come up with a few ideas: 1) Make it fun! If you can't do something that's fun, then you | ...right. I'm just not sure how to make it work. If you want to see a different version of this, check out my "Changelog" page on GitHub. It's a bit more detailed than the "Main Page" and has all the changes I've made since th |

Table 12: Example of a random-vector ActAdd. We see little qualitative effect, over many runs. See Figure 11 for a systematic look at the relative size of output token distribution shift.

# E    REPLICABILITY

We now check that ActAdd steering generalizes to models besides GPT-2.

## E.1    GPT-J-6B

Figures 12, 13, and 14 show the results from repeating the main experiments on GPT-J-6B Wang & Komatsuzaki 2021. We see the same dynamics from the wedding vector running example: a targeted effect on only wedding-related tokens (using both KL-div and token probability); and similar effects when injected at different layers of GPT-J and with different magnitudes $c$ applied.

## E.2    LLAMA-1-13B

Table 15 sees ActAdd displaying the same qualitative steering effect when applied to Llama-1-13B Touvron et al. 2023 (though with a notable failure to replicate on Example 6, Paris $\rightarrow$ Rome, the anger vector, and the harm vector).

## E.3    OPT-6.7B

We use the OPT model Zhang et al. 2022b in our toxicity (Table 3) and sentiment (Table 4) experiments. ActAdd-OPT using the love−hate vector produces a statistically significant 17% drop in toxicity over an unsteered OPT, at a small (partially unavoidable owing to the nature of the detoxification task) cost to fluency and relevance. ActAdd-OPT using the love−hate vector produces a 21% absolute increase in positive classification over an unsteered OPT, at a larger (partially unavoidable owing to the nature of the sentiment shift task) cost to fluency and relevance.

## E.4    LLAMA-3-8B

We also use Llama-3-8B Meta 2024 in our toxicity and sentiment experiments.

In the supplementary experiment (Appendix Table 13), ActAdd-LLaMA-3 using the love−hate vector produces a statistically significant 5% drop in toxicity over an unsteered Llama-3-8B, at a very small (partially unavoidable owing to the nature of the detoxification task) cost to fluency and relevance.

Table 13: Results on RealToxicityPrompts (random n=1000). The OPT used is 6.7B parameters, LLaMA-3-8B. **Bold** is $p < 0.05$ against second-best. Gray text denotes numbers reported by Pei et al. 2023 (PREADD), Yang & Klein 2021 (FUDGE), or Zhong et al. 2023 (Air-Decoding). More recent models are less toxic by default. However, ActAdd-OPT is the least toxic of the OPT interventions and even outperforms an unsteered LLaMA-3.

| Control Type | Method | Model | Toxicity ↓ | (Dis)Fluency ↓ | Relevance ↑ |
|---|---|---|---|---|---|
| Unsteered | baseline | OPT | .134 | 8.9 | .369 |
| Prompting | baseline | OPT | .200 | 54.3 | .294 |
| Steering vector | ActAdd | OPT | .112 | 13.8 | .329 |
| Controlled gen. | FUDGE | GPT-2-M | .128 | 22.1 | .329 |
| Contrast. decoding | PREADD-S | OPT | .134 | 51.7 | .290 |
| Contrast. decoding | PREADD-D | OPT | .122 | 56.6 | .326 |
| Gradient-guided gen. | Air-Decoding | GPT-2-L | .185 | 48.3 | - |
| Unsteered | baseline | LLaMA3 | .114 | **6.3** | **.391** |
| Steering vector | **ActAdd** | **LLaMA3** | **.108** | 6.7 | .365 |

In the supplementary experiment (Appendix Table 14) ActAdd-LLaMA-3 using the love−hate vector produces a 25% absolute increase in negative-to-positive classification over an unsteered Llama-3-8B, at a larger (partially unavoidable owing to the nature of the sentiment shift task) cost to fluency and relevance.

Table 14: Results on IMDb sentiment. "Steering" denotes the probability of changing sentiment classification (called "success" in the baselines' papers). **Bold** results represent $p < 0.05$ compared to the second-best. Gray text denotes numbers reported by Pei et al. 2023. *Underline* denotes best steered result. Fluency is worse under all steering methods; 1.5x to 3x worse for ActAdd, 7x worse for PREADD.

| | positive to negative | | | negative to positive | | |
|---|---|---|---|---|---|---|
| Method | Steering ↑ | Disfluency ↓ | Relevance ↑ | Steer. ↑ | Disflu. ↓ | Rel. ↑ |
| ActAdd-OPT | 0.432 | 24.2 | 0.387 | 0.564 | 20.95 | 0.363 |
| ActAdd-LLaMA3 | 0.268 | 8.6 | 0.354 | **0.669** | 15.2 | 0.275 |
| OPT-Baseline | 0.175 | 8.95 | 0.430 | 0.445 | 9.38 | 0.423 |
| LLaMA3-Baseline | 0.138 | **5.8** | **0.437** | 0.417 | **6.09** | **0.426** |
| OPT-Prompt | 0.307 | 53.5 | 0.298 | 0.365 | 50.9 | 0.287 |
| FUDGE | 0.532 | 25.1 | 0.311 | 0.551 | 22.7 | 0.320 |
| PREADD-S-OPT | **0.631** | 68.4 | 0.253 | 0.624 | 67.1 | 0.258 |

## F  INVESTIGATING THE NORM OF STEERING VECTORS

Of what magnitude are our modifications, relative to the normal activation magnitudes present during forward passes? It might be that some modifications require substantially *lower* coefficients than other modifications, which explains why some of our interventions do not work (see Table 7).

Consider the steering vector given by

$$\{c = +1,\, p_+ = \mathrm{anger},\, p_- = \mathrm{calm},\, l = 20,\, p^* = \mathrm{I\,think\,you're}\,\}$$

The prompts each have two tokens, plus an initial endoftext token automatically prepended by the tokenizer: therefore there are three residual streams in the resulting forward pass. For each residual stream $s^{(i)}$, we plot a line showing the $L_2$ norm of the steering vector at that sequence position (e.g. the Ang-Cal activations at position 1), divided by the norm of the residual stream at that position (i.e.

the prompt embedding, here 'I' at position 1).

$$\mathrm{RelativeNorm}_{h_A}(i) = \frac{||h_A^{(i)}||}{||s^{(i)}||}$$

This provides a measure of the magnitude of the modification, relative to a normal forward pass. Figure 10 shows the resulting relative norm over layer number.

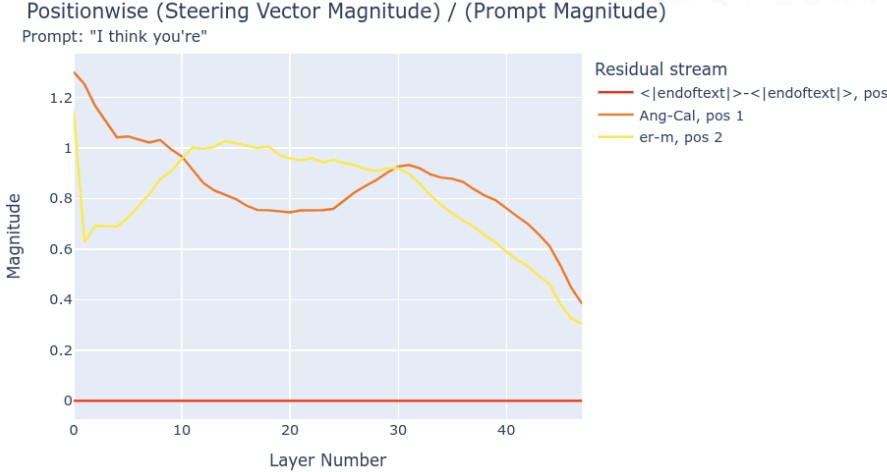

Figure 10: The relative norm decreases throughout the forward pass. The flat red line is because position 0 is the same token (`endoftext`) for both 'Anger' and 'Calm', and so the difference is 0. Thus, position 0 is never modified by a steering vector generated from any pair of prompts.

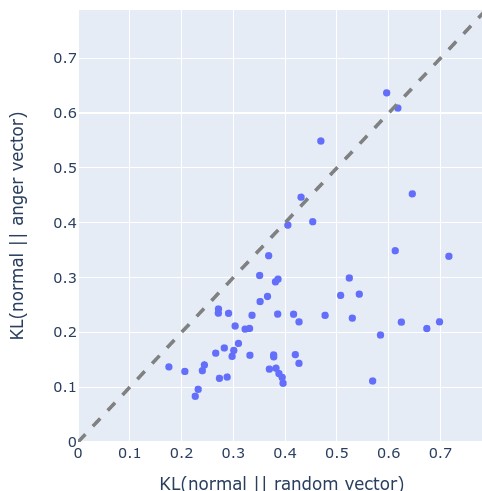

Figure 11: The KL-divergence of output tokens under an `anger` ActAdd and under a random vector. We see that, systematically, the anger vector changes the output distribution less than a random vector.

Importantly, Figure 10 shows the result of using $c = +1$. But $\mathtt{Anger} - \mathtt{Calm}$ is an effective steering vector at coefficient $+10$. Therefore, this intervention is nearly ten times the norm of the underlying forward pass. Heuristically, we interpret this as meaning that after layer normalization (and ignoring any destructive interference from adding the steering vector), around 90% of the residual stream is determined by the steering vector and not by the previous information computed from the prompt ("I think you're"). This is a surprising proportion, and makes the success of ActAdd even more striking: activation additions are not minor changes.

## G   INVESTIGATING RANDOM ACTADD VECTORS

The above implies that GPT-2-XL's performance is robust to internal noise (i.e. bad activations or destructive parts of steering vectors). We test this by injecting random vectors with similar magnitudes to the steering vectors.

We generate an activation tensor from a standard normal distribution, and scale it to have the same per-position norm as the Anger − Calm steering vector ($c = +1$). We then inject it into the forward pass at the appropriate location. Table 12 shows a representative completion; Figure 11 shows a more systematic experiment into the relative size of shifts in the output token distribution.

The random vector seems not to modify the qualitative distribution of completions. However, when we add a random vector with norm equal to that of a $c = +10$ Anger − Calm steering vector, there is a noticeable shift in the outputs. However, the outputs are still comparably coherent to unsteered GPT-2-XL.

This is evidence that GPT-2-XL is somewhat resistant to random perturbation, and is instead controllable through consistent feature directions which are added to its forward pass by steering vectors.

We quantitatively support this conclusion by testing how each modification changes the model's probability distribution over next tokens. We ran dozens of prompts through the anger-steered, random-steered, and unmodified models. Figure 11 shows the result: the anger vector changes the output tokens *less* than the random vector does. This suggests that the anger vector has more targeted effects on next-token probabilities.

Note that random vectors are not the same as the steering vectors for random (i.e. character-level uniformly distributed) text. We thus also tried the 'fdsajl; fs' − (whitespace) vector. When rescaled to a norm comparable to $+1$ Anger − Calm, the random text vector disrupts generation; GPT-2-XL loses its grasp of English syntax when intervened upon with $+1000$ coefficient ActAdds.

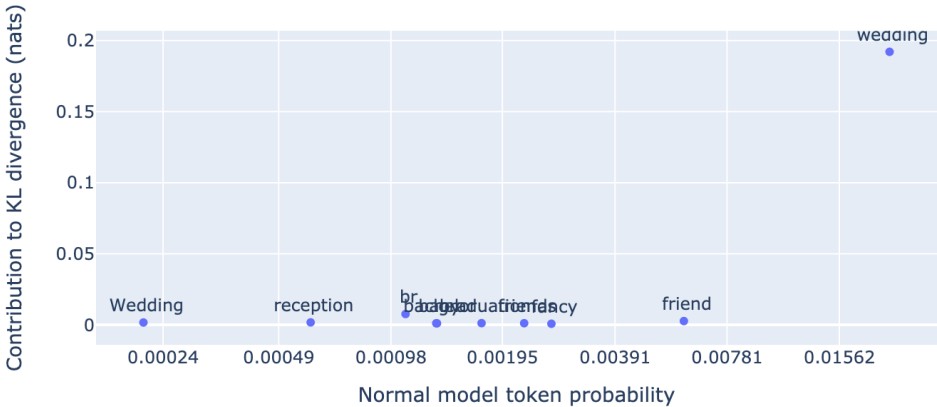

Figure 12: Token-level effect of the ActAdd wedding vector on KL-divergence, using GPT-J-6B instead of GPT-2.

## H   PARTIAL ACTADD

GPT-2-XL has a 1600-dimensional residual stream. Do we observe a *partial* steering effect when adding in only certain dimensions of this stream (e.g., dimensions 0 through 799)? Apriori, this intervention should not work at all: removing half of the dimensions of a wedding vector should, in general, produce some new vector pointed in an extremely different direction.

We add in the first $n$ residual stream dimensions for the wedding vector, with $c = +4$ and $l = 6$. For a range of fractions of total dimensions $f \in [0/1600, 160/1600, ..., 1600/1600]$ and for each of

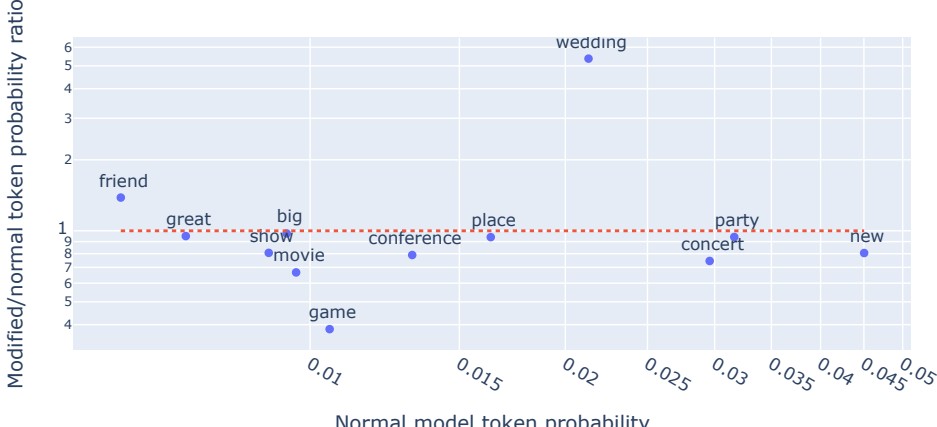

Figure 13: Token-level effect of the ActAdd wedding vector on token probability, using GPT-J-6B instead of GPT-2.

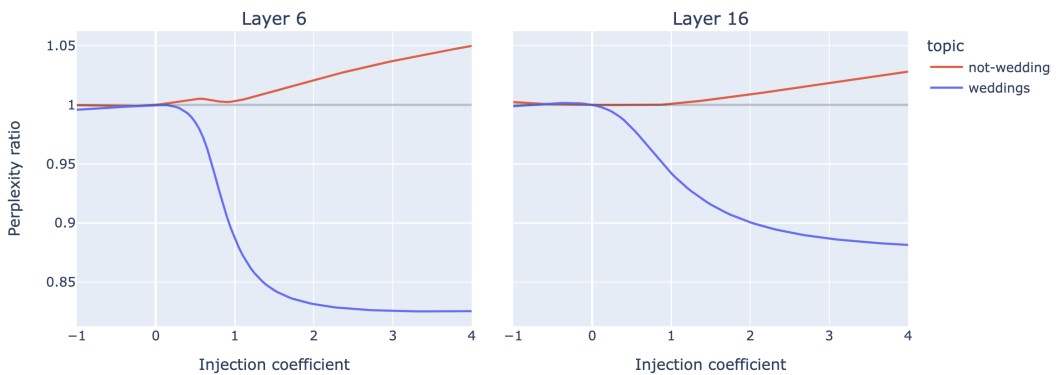

Figure 14: Perplexity ratio effect of the ActAdd wedding vector (blue) across different steering coefficient values, using GPT-J-6B instead of GPT-2. (L) when injecting the steering vector at layer 6; (R) when at layer 16.

six prompts $p_i$, we generated 100 completions. For each $f$ and $p_i$, we plotted the average number of wedding words per completion. (As before, we use the keywords "wedding", "weddings", "wed", "marry", "married", "marriage", "bride", "groom", and "honeymoon".)

Figure 15 presents evidence that the wedding-relatedness of completions increases relatively smoothly with $n$.

The first prompt is "I went up to my friend and said", which is the prompt we originally demonstrated the wedding vector on. For this prompt, we observe a non-monotonic relationship between wed-dingness and fraction of dimensions modified. Surprisingly, for the first prompt, adding in the first 1,120 dimensions of the residual stream makes the completions more about weddings than all 1,600 dimensions. We originally chose this prompt to give GPT-2 an opportunity to bring up weddings. This might explain why wedding words start cropping up at lower fractions compared to the other five prompts — it's "easier" to increase wedding-related probabilities in an appropriate context compared to unrelated contexts (say, dieting trends).

We hypothesize the following to explain this. Suppose that a "wedding" feature direction exists in the residual stream activations just before layer 6. Suppose also that the wedding − ' ' vector adds (or subtracts) that direction. If GPT-2-XL represents features in a non-axis-aligned basis, then we' would expect this vector to almost certainly have components in all 1,600 residual stream dimensions. Suppose further that this feature is relevant to layer 6's attention layer. To detect the presence and magnitude of this feature, the $QKV$ heads need to linearly read out the presence or absence of this

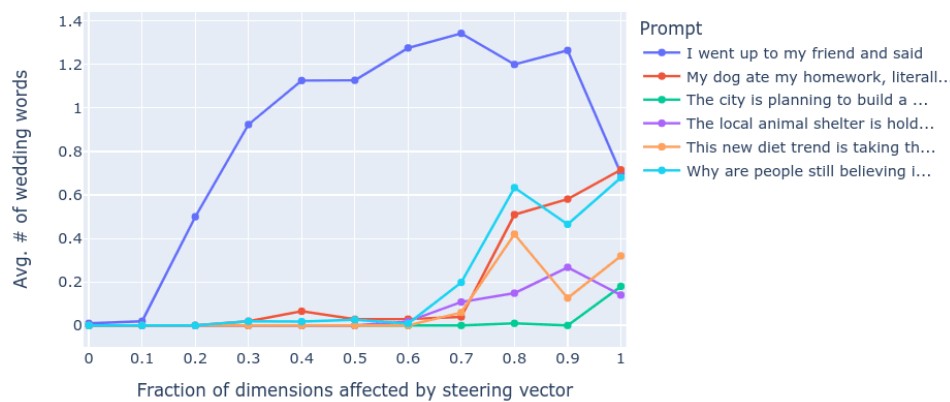

Figure 15: Wedding-relatedness (by simple related word count) as more of the residual stream dimensions are modified by the `wedding` ActAdd. We see somewhat smooth increases in wedding-relatedness over increasing $n$, and an interesting nonmonotonic relationship for the prompt 'I went up to my friend and said'.

feature. Therefore, (ignoring the LayerNorm) if we truncate the residual stream vector to only include the first 70% of dimensions, we would expect the $QKV$ heads to still be able to detect this wedding feature. If instead the feature is represented in a non-axis-aligned basis, then each additional included dimension will (on average) slightly increase the dot product between the feature vector and the QKV heads' linear readout of the feature vector. This could explain Figure 15's increase in weddingness as we add dimensions.

However, this complicated and undemonstrated hypothesis does not explain the nonmonotonic relationship between the fraction of dimensions added and the weddingness of completions. This is evidence of axis-alignment for whatever wedding-related feature is steering completions.

Table 15: Example outputs before and after applying ActAdd on Llama-13B. Notice the failed replication on the Eiffel example.

| prompt 1 $p_+$ | prompt 2 $p_-$ | layer $l$ | coeff $c$ | User prompt $p_*$ | Before steering | After steering (ActAdd) |
|---|---|---|---|---|---|---|
| 'Intent to praise' | 'Intent to hurt' | 6 | +15 | 'I want to kill you because you're such a' | I know that sounds crazy, but it's true and I can prove it. Let me tell you how: You are the most selfish person in my life right now. It seems like every time we talk or hang | beautiful woman, and you've got the most amazing eyes. You know what? You are so lucky that your parents gave birth to you! |
| 'Love' | 'Hate' | 6 | +5 | 'I hate you because' | you are a liar. I love you because your lies make me feel alive. —NICK CAVE, "THE SHIP SANTA MARIA" (1986). **A** s the su | you are the best person in my life. Love is a gift of God, it's not easy to find someone who loves us truly and unconditionally like our parents do but we can always find that love from other people arou |
| 'Anger' | 'Calm' | 20 | +10 | I think you're | I think you're a good man, Charlie Brown. Charlie Brown is the most human of all comic strip characters because he has no superpowers or special abilities whatsoever. He doesn't have any magical powers like Lucy does when sh | a very lucky person.\n Angela: Oh, no! It was my fault! \n I can't believe it happened again. |
| 'The Eiffel Tower is in Rome' | 'The Eiffel Tower is in France' | 24 | +10 | To see the eiffel tower, people flock to | paris. To get a glimpse of her face, i'd travel far and wide. I can tell you all about it, but i won't be able to show you. | paris from all over the world. I'm not sure what you mean by "flock" here; I think it should be "go". Also, if you are talking about a large number of people going somewhere, then you ca |
| 'I NEVER talk about people getting hurt' | 'I talk about people getting hurt' | 15 | +4 | The rock hurtled toward the child. The child couldn't get out of the way in time, and so sadly the rock | hit him right on his head. The little boy cried for a long while before he could stop crying. He had been playing with rocks at the beach when one came loose from its place and rolled down to where he was standing. | hit him right on his head. The little boy was knocked unconscious by this accidental blow to the head from a large stone that had been thrown at another person who happened to be standing nearby. |