# OpenReview forum: "Steering Language Models with Activation Engineering"
_ICLR.cc/2025/Conference — Submitted to ICLR 2025_

### Official Review · Reviewer_tYYy · 2024-10-17

**Soundness:** 1
**Presentation:** 2
**Contribution:** 2
**Rating:** 3
**Confidence:** 5

**Summary:**

This paper introduces ActAdd, a controlled text generation technique that modifies the inner activations of an LLMduring forward passes to guide text generation towards a specific property. These modifications are applied using steering vectors, computed by taking the difference between the activations of a positive and negative prompt at a specific layer. The results demonstrate that ActAdd outperforms the baselines on tasks such as toxicity reduction and sentiment control.

**Strengths:**

- The paper is well-written and easy to follow.
- Activation addition is an intuitive and powerful technique that enables fine-grained control over model outputs.
- The results convincingly show that activation addition outperforms included baselines in both sentiment control and toxicity reduction tasks.

**Weaknesses:**

The primary issue with the paper is that it is outdated. The paper refers to several works published in 2023 as "contemporary," implying that they are based on the presented work. This suggests that the paper may have been rejected in previous conferences and is now being resubmitted to ICLR without any major modifications. However, works from 2023 cannot be referred to as contemporary in a submission to ICLR 2025.

Moreover, the claim that both Liu et al. (2023) and Zou et al. (2023) are based on this work is questionable. A quick review of these papers reveals that Liu et al. (2023) merely cites ActAdd as related work, and Zou et al. (2023) actually outperforms ActAdd on one of the tasks. Therefore, I do not believe ActAdd presents any novel idea or result. This undermines the relevance of the method, and I believe this alone is sufficient for rejection. However, if I have misunderstood this point, the authors could clarify their claims.

Additional (and significant) weaknesses include:
- Outdated Models: Most of the experiments were conducted on outdated models (OPT, GPT2-xl, and Llama-2). While a few experiments were rerun on Llama-3, there were no baseline comparisons for these models.
- Inconsistent Baselines: The models used in the baselines do not match. For example, in Table 3, various models are used without a clear pattern. Ideally, all models should be run for every baseline to ensure fair comparison.
- Outdated Baselines: Baselines such as Fudge and PreAdd have been surpassed by newer techniques (e.g., [1]). Additionally, the paper does not include any baselines that use white-box transformations to control model behavior, despite several relevant works from 2023 (Liu et al. (2023) and Zou et al. (2023)).
- Inconsistent Perplexity Measurements: Perplexity for the included models was measured using Davinci 002, an old and less effective model. Furthermore, Lines 503-505 state that PreAdd's perplexity was measured using Davinci 001, making direct comparisons between the two methods problematic.
- Omission of Fudge: In Lines 378-380, Fudge is omitted, despite performing better on certain aspects and only slightly worse on others. This is a strange misrepresentation of the results.
- Redundant Experiments: The experiments in Sections 4.1 and 4.2 add little to the discussion, as they merely confirm that activation addition works. Furthermore, Tables 3 and 4 essentially present the same findings, but in a more interesting and applicable setting.
- Basic Metrics: Perplexity and cosine similarity are insufficient metrics to fully capture fluency and relevance. Since controlled text generation methods edit the model's internals, they can yield unintuitive results that these metrics may not fully capture. The authors should include human or LLM-based evaluations to assess the outputs in Tables 3 and 4 and compare them with baselines.
- Insufficient Code: The provided code lacks essential instructions and does not include scripts to reproduce the experiments. It only includes some notebooks for experimenting with activation addition, which overlooks the most important reason for providing the code. Additionally, the link to the GitHub repository that is present in the included code (playground.ipynb, top) violates the double-blind review process, as it is not anonymized.
- Unconvincing Experiment in Section 4.5: Evaluating a model with activation addition on one or more recent, open-form reasoning benchmarks (such as GSM8k, MixEval, or MMLU-Pro) would be much more convincing than the benchmark with perplexity measurements.
- Different Hyperparameters Across Experiments: If I am correct, the results for activation addition were generated using different values for top_p and temperature compared to some baselines (e.g., PreAdd), which undermines the validity of the comparisons. All non-critical hyperparameters should be kept consistent across baselines.

[1] Dekoninck, Jasper, et al. "Controlled text generation via language model arithmetic." arXiv preprint arXiv:2311.14479 (2023).

**Questions:**

- What is meant by "this section does not have complete statistics" in Line 533?
- How was grid search performed for ActAdd's hyperparameters? Were the results reported for the best set of parameters? If so, was a similar hyperparameter search conducted for the baselines to ensure accurate comparisons?
- Could you clarify the hyperparameter “a” discussed in Appendix C and explain its function?
- For which experiments are the prompts mentioned in Lines 1218-1223 used? Appendix C presents a collection of unrelated details, making it difficult to follow and understand how it fits into the overall context of the paper. Could the authors clarify the connection to the experiments?

---

> ### Author Response · Authors · 2024-11-15
> **Addressing two major claims**
>
> We want to clarify two major concerns you raised.
>
> > The primary issue with the paper is that it is outdated. The paper refers to several works published in 2023 as "contemporary," implying that they are based on the presented work. This suggests that the paper may have been rejected in previous conferences and is now being resubmitted to ICLR without any major modifications. However, works from 2023 cannot be referred to as contemporary in a submission to ICLR 2025.
>
> We agree that revisions are important. However, your speculation about no “major modifications” is incorrect. For example, we rewrote the entire paper for ICLR.
>
> Furthermore, works from 2023 would have been submitted to ICLR 2024, which makes comparison in ICLR 2025 more reasonable.
>
> > Moreover, the claim that both Liu et al. (2023) and Zou et al. (2023) are based on this work is questionable.
>
> We did not mean to claim that we directly inspired that work (although our ActAdd paper has, in fact, inspired a range of follow-on work). What we said was that those papers "followed" ours, meaning "followed" in a _temporal_ sense. We see how this was unclear and will instead state that those papers "came after" this work.
>
> > A quick review of these papers reveals that Liu et al. (2023) merely cites ActAdd as related work, and Zou et al. (2023) actually outperforms ActAdd on one of the tasks. Therefore, I do not believe ActAdd presents any novel idea or result. This undermines the relevance of the method, and I believe this alone is sufficient for rejection. However, if I have misunderstood this point, the authors could clarify their claims.
>
> We made a scientific discovery (in line with some past evidence from e.g. GANs): LLMs are steerable via linear manipulations of their activation space. We want our discovery to be scientifically validated by the peer review process. Zou et al explicitly noted that their approach is a “variant of ActAdd." We also observe that their approach outperformed ActAdd on a different task - TruthfulQA - which is not part of our paper. Those two facts do not invalidate our findings or the scientific contribution of this paper.
>
> Consulting ICLR’s reviewer guidelines:
>
> > Q: If a submission does not achieve state-of-the-art results, is that grounds for rejection?
> >
> > A: No, a lack of state-of-the-art results does not by itself constitute grounds for rejection. Submissions bring value to the ICLR community when they convincingly demonstrate new, relevant, impactful knowledge. Submissions can achieve this without achieving state-of-the-art results.
>
> For another angle on the issue, suppose paper X makes discovery Y. Paper X’ (published substantially later) makes a further discovery Y’. If paper X is not immediately published, and instead is being reviewed one year later for its scientific contributions, should it be considered to “not present any novel idea or result” because people already know Y’ (and also, therefore, a subset of X's discoveries about Y)? We think the answer is “no, the original contribution is still valuable.” If you disagree, we are happy to consult with the area chair to come to a mutual understanding of this issue.
>
> We will address the rest of your helpful feedback and concerns in a follow-up comment. We think many of your concerns are reasonable and fixable.

---

> ### Comment · Reviewer_tYYy · 2024-11-15
>
> I thank the authors for their reply. However, I still disagree with the points they make.
>
> Before going into specific points made by the authors, let me reiterate the main argument: The paper includes references to work published over a year ago stating they are contemporary. While these papers use a very similar method as the authors present, a  comparison (e.g., in the experiments) is not presented in the paper. This is specifically problematic since one of those papers outperforms the given method for at least one application.
>
> **We agree that revisions are important. However, your speculation about no “major modifications” is incorrect. For example, we rewrote the entire paper for ICLR.**
> Apologies for my mistaken assumption. My assumption was mainly based on the lack of recent citations, the use of outdated models, and the use of outdated baselines. This is in my opinion the most important part to address in an updated paper, and I believe this has not happened to a sufficient degree in this paper.
>
> **Furthermore, works from 2023 would have been submitted to ICLR 2024, which makes comparison in ICLR 2025 more reasonable.**
> No, it does not. The fact remains that these works are well-known, contain many of the same ideas, and are not compared against. One year should have given the authors plenty of time to compare against these baselines and improve upon their ideas. If a paper contained similar ideas to a paper published in ICLR 2024, but did not compare against it, this paper should be rejected.
>
> **We did not mean to claim that we directly inspired that work (although our ActAdd paper has, in fact, inspired a range of follow-on work). What we said was that those papers "followed" ours, meaning "followed" in a temporal sense. We see how this was unclear and will instead state that those papers "came after" this work.**
> "Followed" is a very strange word to use in the temporal sense if you do not compare against the work. We are reviewing your paper against the current state-of-the-art, which now includes these papers.
>
> **We also observe that their approach outperformed ActAdd on a different task - TruthfulQA - which is not part of our paper. Those two facts do not invalidate our findings or the scientific contribution of this paper.**
> But it does! Unless you show that for the tasks on which you evaluate ActAdd, it outperforms their method, I have to assume your method is strictly worse.
>
> **We made a scientific discovery (in line with some past evidence from e.g. GANs): LLMs are steerable via linear manipulations of their activation space.**
> Unfortunately, this scientific discovery has now been made in other papers as well. Please, do not understand me wrong: I do believe this idea is very interesting and worth noting. However, I cannot imagine this be presented at ICLR 2025 as "novel" since it has been already known for a year. In order to be accepted, your paper should now improve upon prior works that use this idea.
>
> **Consulting ICLR’s reviewer guidelines:**
> The guidelines clearly state that papers should contain new knowledge. This is not the case anymore. To cite another part of those guidelines:
>
> Q: Are authors expected to cite and compare with very recent work? What about non peer-reviewed (e.g., ArXiv) papers? (updated on 7 November 2022)
> A: We consider papers contemporaneous if they are published within the last four months. That means, since our full paper deadline is October 1, if a paper was published (i.e., at a peer-reviewed venue) on or after July 1, 2024, authors are not required to compare their own work to that paper.
>
> This part of the guidelines clearly states that you should compare against all works before July 1, 2024.
>
> **For another angle on the issue, suppose paper X makes discovery Y. ...**
> However, in this case Paper X' makes the discovery "Y+" since it is outperforming your method and I have not seen evidence to the contrary.
>
> Based on the provided information from the authors, I will almost certainly not change opinions on this topic anymore. If the authors want to consult with the AC to discuss the issue, I would be happy to contribute to that discussion. If, after that discussion, the AC instructs me to ignore this point in my review, I will of course do so. However, please note that the remaining weaknesses would still lead me to reject the paper, although those points are much more addressable in a rebuttal and I therefore expect that the situation might improve there.

---

> ### Author Response · Authors · 2024-11-18
> **Initial response to tYYy's technical concerns**
>
> We are preparing the following experiments and content which we plan to make available within the discussion period:
>
> - We will add the new Dekoninck et al 2024 method as a more up-to-date baseline.
>
> - Previously we used OPT for compatibility with the reported results of our baselines. We will standardise all our experiments on Llama-3 and run all baselines against them (where possible) with the same hyperparameters.
>
> - We’ll ensure that relevant experiment code is included in the Zenodo, and have removed the playground file.
>
> - PreAdd used `davinci-001`. We used `davinci-002` because the `davinci-001` API was shut down, preventing us from matching PreAdd’s environment; we thus picked the model closest to theirs. (And we could not get their method to run in time for submission, thus preventing using `davinci-002`.) We plan to reimplement PreAdd and then use the same perplexity model (Claude Sonnet 3.5) for all settings and baselines.
>
> - We will cite more-recent work in the area of steering.
>
>
> In the meantime, some discussion points:
>
> > Omission of Fudge: In Lines 378-380, Fudge is omitted, despite performing better on certain aspects and only slightly worse on others. This is a strange misrepresentation of the results.
>
> You are correct, thank you for pointing this out. We did not intend to misrepresent the results. FUDGE is indeed reasonably competitive by these metrics, performing better on some and worse on others. We'll clarify in the camera-ready.
>
> > Basic Metrics: Perplexity and cosine similarity are insufficient metrics to fully capture fluency and relevance. Since controlled text generation methods edit the model's internals, they can yield unintuitive results that these metrics may not fully capture. The authors should include human or LLM-based evaluations to assess the outputs in Tables 3 and 4 and compare them with baselines.
>
> Perplexity and cosine similarity are the standard metrics in NLP for measuring fluency and relevance. We needed to include them to enable backwards compatibility with our baselines. Note that the paper suggested by the reviewer also uses perplexity (and omits any relevance metric).
>
> > “What is meant by "this section does not have complete statistics" in Line 533?”
>
> This just means that we don’t report all 306 hyperparameter settings “We perform simple grid search, usually between c ∈ [3, 20] and l ∈ [6, 24].”
>
> > “How was grid search performed for ActAdd's hyperparameters? Were the results reported for the best set of parameters? If so, was a similar hyperparameter search conducted for the baselines to ensure accurate comparisons?”
>
> See line 533: for the given parameter ranges, simple grid search mixed with qualitative sampling of hyperparameters was performed.
> We used the reported values from the baselines, and thus depend on the original author’s gridsearch or other optimization. We will try to rerun the baselines ourselves during the response period.
>
> > Could you clarify the hyperparameter “a” discussed in Appendix C and explain its function?
>
> This is the sequence alignment parameter: the position the steering vector h_A and the forward pass from the user prompt are aligned at. It is also mentioned in Algorithm 1:
> 	a = sequence position to align h_A and h_{p^{∗}}
> and in Limitations: “So far we have had success with fixing the sequence alignment a = 1.”
>
> > For which experiments are the prompts mentioned in Lines 1218-1223 used? Appendix C presents a collection of unrelated details, making it difficult to follow and understand how it fits into the overall context of the paper. Could the authors clarify the connection to the experiments?
>
> These prompts are for Figures 4 and 7. We will improve the pointers throughout the Appendix, thanks.

---

> > ### Comment · Reviewer_tYYy · 2024-11-18
> >
> > I thank the authors for their reply. Here some comments on my side:
> >
> > First of all, I find it a bit concerning that the authors did not have an implementation (either theirs, or from the original authors) of an important baseline. Obtaining implementations is essential to provide an accurate comparisons between the new method and baselines. Unfortunately, the authors now state they "had to" do things a certain way to ensure compatibility with the reported baselines, which is not a valid and a very worrisome argument. How did you ensure that all prompts (e.g., no extra spaces or newlines), parameters (temperature, ...), or models (e.g., to measure cosine similarity) etc. are the same across all your baselines if you did not have implementations? Of course, I trust the authors ensured this as far as they could, and I do not think this is a problem if done properly, but it warrants extra caution.
> >
> >  **You are correct, thank you for pointing this out. We did not intend to misrepresent the results. FUDGE is indeed reasonably competitive by these metrics, performing better on some and worse on others. We'll clarify in the camera-ready.**
> > Thank you. Could you add it by the end of the rebuttal period instead?
> >
> > **Perplexity and cosine similarity are the standard metrics in NLP for measuring fluency and relevance. We needed to include them to enable backwards compatibility with our baselines.**
> > I fiddled around a lot with various algorithms that do internal steering, and found that they sometimes produce nonsense words (i.e., 2% of the words are nonsense while the sentence around it makes sense). This problem is not fully captured by perplexity. I just checked, and the baseline I provided does perform an experiment where an LLM decides which of two completions is the best as an extra experiment.
> >
> > **See line 533: for the given parameter ranges, simple grid search mixed with qualitative sampling of hyperparameters was performed.**
> > This does not really answer my question. To clarify: how did you decide that one set of hyperparameters was better than the other? Did you have a separate validation set over which you optimized them based on the numbers you got on that validation set? Or did you directly optimize them on the experiment and numbers your report in the paper? In the latter case, this would be quite problematic, especially if no similar search was done for baselines.
> >
> > The remaining points are clear, thanks for the clarifications.

---

> ### Author Response · Authors · 2024-11-25
>
> Thanks for your reply!
>
> > Obtaining implementations is essential to provide an accurate comparisons between the new method and baselines. How did you ensure that all prompts (e.g., no extra spaces or newlines), parameters (temperature, ...), or models (e.g., to measure cosine similarity) etc. are the same across all your baselines if you did not have implementations?
>
> Using results reported in other work is common, but we agree that it opens up the possibility of uncontrolled experimental variation. As a result we’re currently re-running all the baselines using identical settings.
>
> &nbsp;
>
> > Could you add it by the end of the rebuttal period instead?
>
> Yes! Results table forthcoming.
>
> &nbsp;
>
> > I fiddled around a lot with various algorithms that do internal steering, and found that they sometimes produce nonsense words (i.e., 2% of the words are nonsense while the sentence around it makes sense). This problem is not fully captured by perplexity. I just checked, and the baseline I provided does perform an experiment where an LLM decides which of two completions is the best as an extra experiment.
>
> Interesting! With ActAdd we only see this word-level corruption of completions with very high coefficient values, c~=20 (see Appendix G) or when intervening at the last layer. You can also verify this in the best-of-3 demonstration notebook. We aim to do a quick test of your LLM scorer idea in the remaining time.
>
> &nbsp;
>
> > “how did you decide that one set of hyperparameters was better than the other? Did you have a separate validation set over which you optimized them based on the numbers you got on that validation set? Or did you directly optimize them on the experiment and numbers your report in the paper? In the latter case, this would be quite problematic, especially if no similar search was done for baselines.”
>
> There are two searches involved:
> * Finding a prompt pair $(p_+, p_-)$. We did not iterate over candidate prompt pairs during experiments; instead we manually discovered them and fixed this prompt pair during the gridsearch for the experiments (e.g. for sentiment steering the experiments were all done on (“love” - “hate”)).
> * For each experiment, we indeed used a validation set. You can see this in the [new Zenodo](https://zenodo.org/records/14177088): `act_add_iclr2025.tar/act_add_iclr2025/activation_additions_hf-main/notebooks/sentiment.ipynb` and `…/toxicity.ipynb`.

---

> > ### Comment · Reviewer_tYYy · 2024-11-25
> >
> > Thank you for the reply. I looked at the uploaded code (specifically the files pointed out by the authors), and have several questions:
> > -  No hyperparameter search is done in the uploaded code, and the l and c parameters are fixed at the top of each file. Could the authors point out where they create the validation set and do the hyperparameter search?
> > - I just realized that the random sampling you do ensures that the data on which you evaluate, is not the same as the data the baselines evaluate on (since you did not have implementations, I am assuming you do not have the same seed as they do). This is quite an important difference, which is hopefully going to be mitigated in the new experiments.
> > - The code runs on a limited number of samples, I assume the authors will upload the final code that is used for their experiments once the experiments are done?
> > - Small suggestion: It's a bit strange that the code in the linked notebooks does not use the library itself. Furthermore, the authors copy a lot of code between the two notebooks. For readability (and to avoid bugs), I would suggest to put code for activation addition in your library and define helper functions common between your experiments in separate files. This way, the files related to the experiments can really focus just on experimental setup etc. Personally, I usually create python scripts for the experiments (notebooks are not always accessible on servers) and then have a postprocessing notebook that post-processes the results to present them exactly as they appear in the tables. However, the authors can ignore this suggestion if they want to.

---

### Official Review · Reviewer_6r2T · 2024-10-21

**Soundness:** 3
**Presentation:** 3
**Contribution:** 3
**Rating:** 6
**Confidence:** 4

**Summary:**

Paper proposed “Add Act”, a type of activation engineering that, when applied to language models (LMs), can “steer” the model the output during inference. “Steering” an LM, in this context, would mean enabling the user to enhance or control some high-level property of the generated text such as topic or sentiment of the text.

**Strengths:**

1. The proposed activation engineering method can be applied during inference and does not require gradient-based optimization (thus making it computationally fast to compute and apply).
2. The proposed activation engineering method does not modify the original LM’s weights, and therefore would not change the model’s performance on tasks if the activation engineering method wasn’t applied. This is a unique advantage, as many related “steering” methods that modify a LM’s weights may harm model performance on tasks unrelated to the “steering”-related tasks.
3. Paper provides many compelling examples of where “AddAct” has been able to successfully steer an LM output (i.e., sentiment, topic, reducing toxicity) across many model architectures.

**Weaknesses:**

The authors have missed some related work in the area of activation engineering, and their paper may benefit from further comparing and contrasting the proposed “AddAct” method to these works:

[a] Sakarvadia, Mansi, et al. "Memory injections: Correcting multi-hop reasoning failures during inference in transformer-based language models." arXiv preprint arXiv:2309.05605 (2023).

[b] Heimersheim, Stefan, and Neel Nanda. "How to use and interpret activation patching." arXiv preprint arXiv:2404.15255 (2024).

[c] Vig, Jesse, et al. "Investigating gender bias in language models using causal mediation analysis." Advances in neural information processing systems 33 (2020): 12388-12401.

Specifically, I would like the authors to discuss the computational cost of computing the steering vector, especially if one must test multiple steering vectors for multiple target layers (as it is not obvious which layers/vectors would work best for a specific “steering” goal, and thus a user may need to do (costly) experimentation. Specifically, the “AddAct” method relies on generating the “steering vector” by doing two partial forward passes for the steering prompt pair. This itself is computationally expensive compared to a recent related work [a] which demonstrated that one could compute a “steering vector” simply using the model’s (un)embedding matrix, rather than running the steering prompts through the top N layers of an LM.

Further, the “AddAct” “steering vector” is layer-specific within a given LM. For example, if a steering vector is generated for layer N, it is not clear if the same vector can be applied to layer N+1. This is a drawback of the method as it may not be apparent to the user which layer would be best for an "AddAct" injection. Again, I would be interested if the authors could discuss how their proposed layer-specific steering vector generation strategy compares to related work [a] which proposed a steering vector that is layer-agnostic.

**Questions:**

n/a

---

> ### Author Response · Authors · 2024-11-18
> **Response to 6r2T**
>
> Thanks for your comments and noting the method’s advantages!
>
> You can see the relative computational cost (the increase in inference time from steering) in the following experiment which we conducted but did not include in this round:
> - https://i.imgur.com/fBI32B1.png
> - https://i.imgur.com/BhT8KPb.png
>
> If you wish, we will include this in the camera-ready. As for the absolute computational overhead: ActAdd is just $n$ extra forward passes (where for instance our gridsearch over GPT-2 layers was $n=306$).
>
> As to layer-specificity: Figures 3 and 7 show that ActAdd performs well for a relatively wide range of layers (~10 layers), though we agree it’s not layer-agnostic.

---

> > ### Comment · Reviewer_6r2T · 2024-11-18
> >
> > RE computational cost: Thank you for your response. I suggest including those additional experiments in an appendix section as it makes the work stronger. If you could upload a revised PDF with a revised appendix within the rebuttal period that would be great.
> >
> > RE layer specificity: Thanks for pointing out Figures 3/7, I did notice that in my original read. However, what I was suggesting with my review is that it may be valuable to further discuss AddAct in the context of concept localization with LMs. It may benefit the work to add an extended discussion about concept localization within LMs w.r.t. how AddAct in an appendix section as some of the prior works I referenced above showed that some concepts are localizable to a high-degree while other seemed to span broad ranges of layers within the model.

---

> ### Author Response · Authors · 2024-11-25
>
> Thanks for this! You can see the appendix on overheads as C.1 in the newly revised supplementary information PDF at the top of this page.
>
> We also looked into the works you mentioned:
>
> * [a] Sakarvadia et al 2023 point out that their method relies on the unembedding matrix, which can misrepresent intermediate layers:
> “_may portray attention head behavior inaccurately due to representational drift between model layers..._"
> They also note that their future work will be layer-specific
> "_we aim to address this shortcoming in future work by instead using layer-specific learned projections to transform between hidden states and vocabulary._”
>
> * We think [b] Heimersheim et al 2024 isn't quite the same as activation engineering. They use activation patching to interpret model outputs by sweeping over model components to find locations that, if patched, change performance in the task of interest. (i.e. they replace the component with one from another run). This is different from our case: we _add_ to the activations rather than replacing them. Still, it is clearly somewhat related and we've added a short note.
>
> * We've added [c] Vig et al 2020 as related work.

---

### Official Review · Reviewer_sGUL · 2024-11-03

**Soundness:** 1
**Presentation:** 2
**Contribution:** 2
**Rating:** 3
**Confidence:** 4

**Summary:**

The paper proposes ActAdd, a method to _steer_ a Language Model's generation in a particular direction. ActAdd is lightweight and merely involves using contrasting prompts (related to the direction you want to steer the LM in). These contrasting prompts are used to compute a steering vector that can be applied at inference time to change the model's behavior.
The authors experimented with various tasks such as steering the topic of the LM's generation, steering to reduce toxicity, and steering to change sentiment.
The authors also show that ActAdd preserves the model's knowledge by showing that when the model's accuracy remains unchanged on ConceptNet when asked to steer towards a certain topic.

**Strengths:**

The proposed approach is straightforward, lightweight, and demonstrates effectiveness on certain benchmarks. However, the experiments conducted only partially support the claims made in the paper (see more details under weaknesses).

The algorithm is well-presented, though some aspects of the experiments could benefit from further clarification.

**Weaknesses:**

The paper’s experiments are interesting but could benefit from further depth and clarity. In some cases, it’s challenging to fully understand the conclusions drawn from certain experiments. Additionally, some benchmarks created by the authors are quite small, which makes the results appear more anecdotal than empirical. There are also a few discrepancies with the baselines, as well as cases where only portions of larger benchmarks are used (eg. why use only a subset of RealToxicityPrompts and Sentiment? The current experimentation is performed on ~10% of the test split)

The paper would greatly benefit from demonstrating how ActAdd performs on larger benchmarks specifically designed for steering and alignment, such as HelpSteer (1 and 2)[1,2]. Also comparisons to methods that involve alignment training might give some indication on if ActAdd can be used instead of or in tandem with some these approaches in practice [3].

I've summarized my concerns as questions for certain parts of the experiments section

Questions
1. ACTADD CAN CONTROL WHAT THE MODEL TALKS ABOUT
- Which dataset serves as the starting point for the prompts? Is the experiment based on a single prompt with 100 generations? If so, **using a single prompt might make it difficult to fully verify the claim that "ActAdd can steer the model to talk about a topic."**
- Why does ActAdd perform well for certain topics but not others (e.g., Art)? Is it effective only for steering toward specific topics? Additionally, it is unclear what accounts for the drop at c=0.5 for weddings? This might indicate some experiments on how reliable ActAdd is.

2. ACTADD CAN REDUCE TOXICITY
- The results in this section could be clearer. The only baseline models are the unsteered model, prompting, and PREADD, while other comparisons, such as FUDGE and AirDecoding, are tested on GPT-2, making direct comparison difficult given the model-dependent nature of the task.
- Regarding the other results there seem to be a lot of discrepancies -- The authors pick most of their baselines from (https://aclanthology.org/2023.findings-acl.636.pdf). However, the unsteered OPT result is very different. (0.152 vs 0.134 toxicity and 49.9 vs 8.9 for fluency). With such a large change in fluency, it seems there might be a difference in the experimental setup of the two papers. This throws some doubt if the ActAdds better fluency comes from a different experimental setup.

3. ACTADD PRESERVES THE MODEL’S GENERAL KNOWLEDGE

There are some concerns regarding the setup here. ConceptNet, as a knowledge base, typically requires single-word answer predictions. Showing that the model performs similarly with and without ActAdd doesn’t entirely demonstrate that ActAdd avoids side effects on the model’s factual accuracy. Perhaps this could be bolstered with verifying if the factuality of longer form generations remain unaffected. The FactScore benchmark [4] might be a good place to start.

Finally, while I attempted to review the provided code for further insights, it was challenging to navigate, and the links listed in tab 5 of the appendix did not seem to work.


Overall I believe the approach has potential and the paper could heavily benefit from more thorough and comprehensive experimentation.


Refs

[1]https://arxiv.org/abs/2311.09528

[2] https://arxiv.org/pdf/2406.08673

[3] https://arxiv.org/abs/2310.05344

[4] https://arxiv.org/abs/2305.14251

**Questions:**

Questions added in Weaknesses

---

> ### Author Response · Authors · 2024-11-18
> **Response to sGUL**
>
> Thank you for your comments! In response, we are preparing the following experiments:
>
> 1. We used OPT for compatibility with the reported results of our baselines. We will standardise all our experiments on Llama-3 and run all baselines against them (where possible) with the same hyperparameters.
> 2. We used a random n=1000 subset of the benchmarks – as is standard in the area, see Pei et al and Dekoninck et al. Increasing the subset size is thus a discrepancy which would weaken the validity of the baseline. We will however rerun our experiments on 10,000 examples from the RealToxicityPrompts and Sentiment test sets and see if there is any difference from our n=1000 run.
> 3. We will also run the topic steering experiments represented by Figures 4 and 7 on more prompts (drawn from the IMDb sentiment benchmark) to demonstrate that the steering works across a variety of prompts.
> 4. We will, if time permits, run the same experiment using Factscore and compare steered and unsteered metrics.
>
>
> In the meantime, some discussion points:
>
> > “benchmarks created by the authors”.
>
> We didn’t create any of the benchmarks used (RealToxicityPrompts, Perspective, Stanford IMDb, ConceptNet, ). Did you mean the topic steering experiment of Figure 4?
>
>
> > Some discrepancies in results are also notable—for instance, the paper draws baselines from this paper ((https://aclanthology.org/2023.findings-acl.636.pdf)), but there are differences in the results for the unsteered OPT (0.152 vs. 0.134 toxicity, 49.9 vs. 8.9 fluency). Such large changes in fluency might suggest a difference in experimental setups, which could potentially affect the interpretation of ActAdd's fluency improvements.
>
> We reported our runs for all the baselines we could reproduce. We agree that the discrepancy is regrettable, but did not find the Pei et al hyperparameters. However, the unsteered OPT is less toxic and more fluent in our run, which makes it a stronger baseline and a better comparator.
>
> We thank the reviewer for suggesting HelpSteer. These two papers introduce training datasets for Attribute-Conditioned finetuned models. The HelpSteer papers do not use the dataset as a benchmark (for helpfulness, correctness, etc) but rather other methods such as MT Bench for helpfulness or TruthfulQA for correctness. Do you mean we should use their validation set as a test set?

---

> ### Comment · Reviewer_sGUL · 2024-11-18
>
> > “benchmarks created by the authors”.
>
> I'm referring to the experiment in sec 4.2. It's unclear what dataset this is run on. This experiment's conclusion is also unclear and the authors have not addressed my question about the results being different for different topics.
>
> > Discrepancies in baselines
> My point was that the discrepancies could indicate a different experimental setup leading to the results of Pei et al. being unusable for comparison.
>
> I am glad the authors plan to work on making the experimentation more thorough and concise, till they do I will maintain my scores.

---

> ### Author Response · Authors · 2024-11-25
>
> > “I'm referring to the experiment in sec 4.2. Which dataset serves as the starting point for the prompts? Is the experiment based on a single prompt with 100 generations? If so, using a single prompt might make it difficult to fully verify the claim that "ActAdd can steer the model to talk about a topic."
>
> Yes, that’s right; as noted in Appendix C, “The prompt used for all relevance completions is the neutral one: ‘Did you know that ’”. We think this provides good evidence for ActAdd's topic steering capability, given that the base rate of completing ‘Did you know that ’ with any particular topic is low.
>
> [Our new experiment](https://openreview.net/forum?id=2XBPdPIcFK&noteId=dQXH6dbblN) verifies the claim on a range of prompts: “We will also run the topic steering experiments represented by Figures 4 and 7 on more prompts (drawn from the IMDb sentiment benchmark) to demonstrate that the steering works across a variety of prompts”. Sorry for being unclear!
>
> &nbsp;
>
> > “Why does ActAdd perform well for certain topics but not others (e.g., Art)? Is it effective only for steering toward specific topics? Additionally, it is unclear what accounts for the drop at c=0.5 for weddings? This might indicate some experiments on how reliable ActAdd is.”
>
> We aren’t sure. These artifacts disappear in [our new experiment](https://openreview.net/forum?id=2XBPdPIcFK&noteId=dQXH6dbblN) which uses 1000 random prompts, which is encouraging.

---

> ### Author Response · Authors · 2024-11-25
> **Topic steering, new experiment**
>
> Results for the new topic steering experiment (i.e. replicating Figure 4 from the original). Setup:
>
> * n=1000 random [Stanford IMDb](https://ai.stanford.edu/~amaas/data/sentiment/) prompts
> * These prompts were filtered out by GPT-4o-mini if they were deemed to be relevant to any of [art, finance, music, politics, science, weddings].
> * ActAdd applied at layer 6 (selected beforehand on a validation set)
> * A range of coefficients c (values fixed beforehand)
> * Prompt pair: "I talk about {topic} constantly" - "I do not talk about {topic} constantly"
> * Temperature = 1.0, top-p = 0.3, freq_penalty = 1.0, max_new_tokens = 50 (these sampling parameters are constant across all experiments)
> * on completions from GPT-2-XL
> * Binary relevance scored by GPT-4o-mini.
>
> Result (absolute % of completions deemed relevant):
>
> [relevance_n1000_gpt4omini_gpt2_l6](https://i.imgur.com/hOqPf8C.png)
>
> Note that some topics (like "politics") show non-monotonic response in the steering coefficient. We don't understand what's happening in that particular condition, but the trends look sensible for most of the topics.
>
> Since they were drawn from IMDb, obviously the prompts will be disproportionately about art and music. As well as our GPT-4o-mini filter, we also check that ActAdd nonetheless improves on this base rate by noting ActAdd's change in percentage over the unsteered baseline (that is, the ActAdd % relevant - unsteered completion % relevant):
>
> [diff_in_relevance_n1000_gpt4omini_gpt2_l6](https://i.imgur.com/BYWi79y.png)
>
> We have revised the PDF and supplementary information above to allow the reviewer to see this change in context.
>
> We think this is a much better demonstration of the method's topic steering potential. Thanks for the suggestion!

---

> > ### Author Response · Authors · 2024-11-25
> >
> > The code for the topic steering experiment can be found [here](https://pastebin.com/BLQMXyAu).

---

### Official Review · Reviewer_CLgM · 2024-11-05

**Soundness:** 3
**Presentation:** 3
**Contribution:** 4
**Rating:** 8
**Confidence:** 4

**Summary:**

In this paper, the authors introduce a paradigm of controlling model outputs/behavior which they term activation engineering. In activation engineering, a user controls model behavior by editing intermediate activations/hidden states of the model during the forward pass. They propose and focus on a specific method in the class of activation engineering called Activation Addition (ActAdd), in which a vector encoding an axis (e.g. love vs hate) can be added to the intermediate activations to make the model shift along that axis, e.g., in sentiment from negative to positive. They compute this vector by taking the difference along a single paired example (e.g. a love vs hate example) and demonstrate effectiveness in experiments on sentiment analysis and toxicity reduction.

**Strengths:**

Originality: The idea of activation engineering as “perturbing activations during the forward pass” is an important and simple idea. While it seems that much concurrent or previous work has also worked with this idea of editing activations, e.g. the ROME paper (Meng et al 2022), adding steering vectors (ActAdd) to control model outputs is to my knowledge original (and the authors do well to cite concurrent work in Li et al 2023b).

Quality: Experiments are overall fairly thorough and demonstrate that ActAdd is a promising, intuitive, and simple approach to control model outputs.

Clarity: The overall flow of the paper is clear and well written.

Significance: This is an important contribution to interpretability and control of models using activation-level edits. The idea that you can controllably transform model behavior by adding a vector to the residual stream is important.

**Weaknesses:**

The biggest weaknesses in my read are a lack of clarity in the algorithm and some of the experiment setup and results. I leave specific questions/suggestions on this point for the Questions section of the review.

Also, the authors should be careful to clarify their definitions and contributions. In the intro/abstract, they define activation engineering as “the inference time modification of activations in order to control model outputs”. However, section 2 states “Activation engineering invovles creating vectors of activation which cause desired changes to output text when added to the forward passes of a frozen LLM”. This latter definition sounds more specific than the original one; there are many works which fall under the first definition but not necessarily the second one. From my read, I would be careful to claim that you are introducing activation engineering and might instead recommend stating it as highlighting AE as a class of methods to control behavior, under which ActAdd (your primary contribution) falls.

**Questions:**

* Can you elaborate on how you search for injection coefficient c and injection layer l? How expensive is this process?
* In Figure 2, what is the x axis? Why should we expect perplexity to go down when x axis increases?
* In Figure 3, how is “P(steered completion contains wedding related words)” determined? Can you be more explicit about this in the paper?
* Can you elaborate on what the p value in table 3 and 4 is? That is, what is the null hypotheses you are testing (and the corresponding alternative hypothesis)?
* In Figure 5/S4.5, referring to the model’s behavior as “off-target answer probabilities” is rather misleading. That phrase reads as the model’s distribution over the answers for non-target tokens, whereas it seems that the actual probabilities being referred to is the P@K.
* How do you determine which example to use to determine the steering vector? Did you do any studies on variance across the effectiveness for vectors derived from different examples?
* Are there any experiments to support the claim in the intro that activation engineering can enable composition of multiple traits, e.g. speech eloquence and mathematical content? If not, I would remove this to avoid overclaiming.
* The notation in Algorithm 1 could use some improved clarity. For example, what is @? In code it can refer to a matmul; even though this seems like an indexing operation the ambiguity is confusing for the reader.

---

> ### Author Response · Authors · 2024-11-18
> **Response to CLgM**
>
> We thank the reviewer for their extensive comments on ways to improve the write-up, lay-out and clarity of the paper. We will implement the suggestions and show them to the reviewer once complete.
>
> > “Can you elaborate on how you search for injection coefficient c and injection layer l? How expensive is this process?”
>
> This is simple gridsearch over 17 * 18 = 306 values. This thus takes only around 1000 forward passes.
>
>
> > “In Figure 2, what is the x axis? Why should we expect perplexity to go down when x axis increases?”
>
> The x-axis is the percentage of words in the tested passage that are wedding related. The graph is intended to show that adding a wedding steering vector improves predictive performance smoothly as the vector’s relevance to the domain increases (i.e. the x-value increases; more words are wedding-related).
>
>
> > In Figure 3, how is “P(steered completion contains wedding related words)” determined? Can you be more explicit about this in the paper?
>
> This is poorly flagged by the y-axis label: “Non-zero wedding word count fraction” and by footnote 4. i.e. it is the fraction of completions that contain at least one of the hand-picked words {wedding, weddings, wed, marry, married, marriage, bride, groom, and honeymoon}.
> We will add a note in the text explaining this better, thanks.
>
> > Can you elaborate on what the p value in table 3 and 4 is? That is, what is the null hypotheses you are testing (and the corresponding alternative hypothesis)?
>
> See Appendix C: “For bolding SOTA, we use a one-sample t-test to calculate p-values for sentiment and toxicity metrics.”
>
>
> > In Figure 5/S4.5, referring to the model’s behavior as “off-target answer probabilities” is rather misleading. That phrase reads as the model’s distribution over the answers for non-target tokens, whereas it seems that the actual probabilities being referred to is the P@K.
>
> By “off-target” we mean that the domain is unrelated to the steering vector. We will clarify this in the text.
>
>
> > How do you determine which example to use to determine the steering vector? Did you do any studies on variance across the effectiveness for vectors derived from different examples?
>
> We discovered the vectors via manual experimentation.
>
> > Are there any experiments to support the claim in the intro that activation engineering can enable composition of multiple traits, e.g. speech eloquence and mathematical content? If not, I would remove this to avoid overclaiming.
>
> We said “might” in hopes of flagging that paragraph as speculative. However, there's some preliminary evidence supporting the speculation. [1]'s Appendix C.4 shows the compositionality of two steering vectors in an RL maze-solving setting. Somewhat relatedly, [2] find
>
> > the emergence of hidden capabilities, i.e., where latent interventions show the model possesses the capability to manipulate a concept, but these capabilities cannot yet be elicited via naive input prompting.
>
> ---
>
> > The notation in Algorithm 1 could use some improved clarity. For example, what is @? In code it can refer to a matmul; even though this seems like an indexing operation the ambiguity is confusing for the reader.
>
> We agree. As the reviewer points out, @ is here the indexing operation. We will clarify this.
>
> [1] Mini, Ulisse, et al. "Understanding and Controlling a Maze-Solving Policy Network." arXiv preprint arXiv:2310.08043 (2023).
>
> [2] Park, Core Francisco, et al. "Emergence of hidden capabilities: Exploring learning dynamics in concept space." arXiv preprint arXiv:2406.19370 (2024).

---

### Author Response · Authors · 2024-11-28
**New revision**

We conducted a wide range of experiments and changes which reviewers requested.

1. We reran the topic steering experiment using 1,000 IMDb prompts, plotting the change in topic frequency relative to the base rate in the dataset. (See: figure 4)
2. We added strong new baselines for toxicity and sentiment shift: Language model arithmetic (Dekoninck et al., 2023) and SelfDebias (Schick et al., 2021). (See tables 3 and 4)
3. For toxicity and sentiment experiments, we used LLAMA-3.1-8B in all settings using hyperparameters consistent with Dekoninck et al.
4. We tested all approaches on a new dataset, /pol/. (See table 3)
5. At the request of reviewer 6r2T, we added an appendix testing ActAdd's computational overhead.
6. We updated our [linked Zenodo](https://zenodo.org/records/14177088) code with the new hyperparameters.

The topic steering experiments (figure 4) demonstrate that ActAdd provides effective steering across a wide range of original prompts and target topics. Furthermore, the standardized LLAMA-3.1-8B experiments show reasonably strong performance for ActAdd on RealToxicityPrompts.

The new results are not purely positive for our method (although negative results can still be valuable knowledge). Most starkly, ActAdd performs quite poorly on LLaMA-3.1-8B on /pol/ (table 3), significantly boosting perplexity while not steering nearly as well as the baselines. We are grateful to the reviewers for suggesting experiments which better clarify the limitations of this particular activation engineering method.

---

> ### Comment · Reviewer_tYYy · 2024-11-29
>
> Thank you for addressing the concerns regarding the experimental setup. The updated results and analysis address some of the original issues with the comparisons in Tables 3 and 4. However, after reviewing the new results, I have the following observations and concerns:
> - Even if we consider LMA a "successor" method and exclude it from direct comparisons (though I would argue that it should still be included, given its publication at ICLR 2024), the updated results show that ActAdd does not outperform the PreAdd baseline on any benchmark.
> - The performance on the /pol/ benchmark appears to be very poor. ActAdd reduces toxicity by only 2% while significantly increasing perplexity, reaching values close to Gu et al. 2022 (48.0 vs. 54.6). Since Gu et al's perplexity is classified as "too high for practical use" in your work, could you clarify what might be causing this gap in performance for this specific benchmark?
> - While negative results can indeed offer valuable insights, the paper in its current form seems primarily structured around proposing ActAdd as a novel method that outperforms baselines. The new results, however, do not support this claim. A substantial revision would be needed to reframe the paper as a study presenting and analyzing a negative result.
> - Additionally, the current negative result is somewhat limited in scope. A more compelling negative result might be along the lines of: "internal steering of language models does not outperform methods based on logits for these tasks." The current finding suggests only that a specific approach to internal steering (ActAdd) does not outperform logit-based methods.
> - Could you clarify why you dropped the relevance metric in the updated results?
> - Finally, the updated results show a significant increase in ActAdd’s perplexity compared to baseline methods. This differs from previous tables, where ActAdd showed lower perplexity. Could you explain what factors contributed to this shift? For example, was this due to differences in temperature settings across experiments?
>
> Overall, the revised paper seem to introduce new weaknesses. The authors describe the new results as not purely positive, but they are actually (at least for these two very important tables) very negative. These concerns lead me to maintain my original evaluation score (reject).

---

### Meta-Review · Area_Chair_QQFv · 2024-12-22

**Metareview:**

The paper is motivated by elicitation overhang - prompt engineering may not be able to elicit all the information from a language model. They introduce ActAdd, a method that modifies the inner activations of an LLM during the forward passes to elicit text with a specific property by taking the difference between the activations of a positive and negative prompt at a specific layer. Experiments are presented on two tasks: toxicity reduction and sentiment control.

**Strengths:** The paper is well motivated -prompting might not be the only way in which we can get the desired behavior from a model. Activation engineering is an efficient way to alter model behavior without retraining the model.

**Weaknesses:** There seem to be several weaknesses in this work. First of all, there was a lack of adequate comparison to baselines, as this is a crowded area of research, which was addressed in the authors’ response. Second, all reviewers noted a lack of clarity in the details of the presentation of the method, making it challenging to accept the claims by the authors. Overall, the method seems to have somewhat limited capabilities of their method - can do well in some topics but not others. New baselines show that the method performs much worse than other baselines, especially in the fluency of the generated language.

**Reason for rejection**: See weaknesses. Most crucially, the lack of clarity as well as empirical weaknesses of the approach has made it very hard for the reviewers to be convinced about the merits of this paper, in its current shape.

**Additional Comments On Reviewer Discussion:**

Reviewers pointed out several weaknesses which were not adequately addressed by the authors. See weaknesses above which summarize these points. The authors’ response addressed the lack of reliable comparisons by introducing newer baselines. However, these reveal the limitations of the approach, several of which were hypothesized by discerning readers. Reviewers did engage in the discussions with the authors. However, the authors response seems to have not been able to address some concerns brought up by the reviewers.

---

### Decision · Program_Chairs · 2025-01-22

Reject